# PROXIMAL GRADIENT DESCENT-ASCENT: VARIABLE CONVERGENCE UNDER KŁ GEOMETRY

**Ziyi Chen, Yi Zhou**
Department of ECE
University of Utah
Salt Lake City, UT 84112, USA
{u1276972,yi.zhou}@utah.edu

**Tengyu Xu, Yingbin Liang**
Department of ECE
The Ohio State University
Columbus, OH 43210, USA
{xu.3260,liang.889}@osu.edu

## ABSTRACT

The gradient descent-ascent (GDA) algorithm has been widely applied to solve minimax optimization problems. In order to achieve convergent policy parameters for minimax optimization, it is important that GDA generates convergent variable sequences rather than convergent sequences of function values or gradient norms. However, the variable convergence of GDA has been proved only under convexity geometries, and there lacks understanding for general nonconvex minimax optimization. This paper fills such a gap by studying the convergence of a more general proximal-GDA for regularized nonconvex-strongly-concave minimax optimization. Specifically, we show that proximal-GDA admits a novel Lyapunov function, which monotonically decreases in the minimax optimization process and drives the variable sequence to a critical point. By leveraging this Lyapunov function and the KŁ geometry that parameterizes the local geometries of general nonconvex functions, we formally establish the variable convergence of proximal-GDA to a critical point $x^*$, i.e., $x_t \to x^*, y_t \to y^*(x^*)$. Furthermore, over the full spectrum of the KŁ-parameterized geometry, we show that proximal-GDA achieves different types of convergence rates ranging from sublinear convergence up to finite-step convergence, depending on the geometry associated with the KŁ parameter. This is the first theoretical result on the variable convergence for nonconvex minimax optimization.

## 1 INTRODUCTION

Minimax optimization is a classical optimization framework that has been widely applied in various modern machine learning applications, including game theory Ferreira et al. (2012), generative adversarial networks (GANs) Goodfellow et al. (2014), adversarial training Sinha et al. (2017), reinforcement learning Qiu et al. (2020), imitation learning Ho and Ermon (2016); Song et al. (2018), etc. A typical minimax optimization problem is shown below, where $f$ is a differentiable function.

$$\min_{x \in \mathcal{X}} \max_{y \in \mathcal{Y}} f(x,y).$$

A popular algorithm for solving the above minimax problem is gradient descent-ascent (GDA), which performs a gradient descent update on the variable $x$ and a gradient ascent update on the variable $y$ alternatively in each iteration. Under the alternation between descent and ascent updates, it is much desired that GDA generates sequences of variables that *converge* to a certain optimal point, i.e., the minimax players obtain convergent optimal policies. In the existing literature, many studies have established the convergence of GDA-type algorithms under various global geometries of the objective function, e.g., convex-concave geometry ($f$ is convex in $x$ and concave in $y$) Nedić and Ozdaglar (2009), bi-linear geometry Neumann (1928); Robinson (1951) and Polyak-Łojasiewicz (PŁ) geometry Nouiehed et al. (2019); Yang et al. (2020). Some other work studied GDA under stronger global geometric conditions of $f$ such as convex-strongly-concave geometry Du and Hu (2019) and strongly-convex-strongly-concave geometry Mokhtari et al. (2020); Zhang and Wang (2020), under which GDA is shown to generate convergent variable sequences. However, these special global function geometries do not hold for modern machine learning problems that usually have complex models and nonconvex geometry.

Recently, many studies characterized the convergence of GDA in nonconvex minimax optimization, where the objective function is nonconvex in $x$. Specifically, Lin et al. (2020); Nouiehed et al. (2019); Xu et al. (2020b); Boţ and Böhm (2020) studied the convergence of GDA in the nonconvex-concave setting and Lin et al. (2020); Xu et al. (2020b) studied the nonconvex-strongly-concave setting. In these general nonconvex settings, it has been shown that GDA converges to a certain stationary point at a sublinear rate, i.e., $\|G(x_t)\| \le t^{-\alpha}$ for some $\alpha > 0$, where $G(x_t)$ corresponds to a certain notion of gradient. Although such a gradient convergence result implies the stability of the algorithm, namely, $\lim_{t \to \infty} \|x_{t+1} - x_t\| = 0$, it does not guarantee the convergence of the variable sequences $\{x_t\}_t, \{y_t\}_t$ generated by GDA. So far, the variable convergence of GDA has not been established for nonconvex problems, but only under (strongly) convex function geometries that are mentioned previously Du and Hu (2019); Mokhtari et al. (2020); Zhang and Wang (2020). Therefore, we want to ask the following fundamental question:

- *Q1: Does GDA have guaranteed* **variable convergence** *in nonconvex minimax optimization? If so, where do they converge to?*

In fact, proving the variable convergence of GDA in the nonconvex setting is highly nontrivial due to the following reasons: 1) the algorithm alternates between a minimization step and a maximization step; 2) It is well understood that strong global function geometry leads to the convergence of GDA. However, in general nonconvex setting, the objective functions typically do not have an amenable global geometry. Instead, they may satisfy different types of local geometries around the critical points. Hence, it is natural and much desired to exploit the local geometries of functions in analyzing the convergence of GDA. The Kurdyka-Łojasiewicz (KŁ) geometry provides a broad characterization of such local geometries for nonconvex functions.

The Kurdyka-Łojasiewicz (KŁ) geometry (see Section 2 for details) Bolte et al. (2007; 2014) parameterizes a broad spectrum of the *local nonconvex geometries* and has been shown to hold for a broad class of practical functions. Moreover, it also generalizes other global geometries such as strong convexity and PŁ geometry. In the existing literature, the KŁ geometry has been exploited extensively to analyze the convergence rate of various gradient-based algorithms in nonconvex optimization, e.g., gradient descent Attouch and Bolte (2009); Li et al. (2017) and its accelerated version Zhou et al. (2020) as well as the distributed version Zhou et al. (2016a). Hence, we are highly motivated to study the convergence rate of **variable convergence** of GDA in nonconvex minimax optimization under the KŁ geometry. In particular, we want to address the following question:

- *Q2: How does the local function geometry captured by the KŁ parameter affects the variable convergence rate of GDA?*

In this paper, we provide comprehensive answers to these questions. We develop a new analysis framework to study the variable convergence of GDA in nonconvex-strongly-concave minimax optimization under the KŁ geometry. We also characterize the convergence rates of GDA in the full spectrum of the parameterization of the KŁ geometry.

## 1.1 OUR CONTRIBUTIONS

We consider the following regularized nonconvex-strongly-concave minimax optimization problem

$$\min_{x \in \mathbb{R}^m} \max_{y \in \mathcal{Y}} f(x, y) + g(x) - h(y), \tag{P}$$

where $f$ is a differentiable and nonconvex-strongly-concave function, $g$ is a general nonconvex regularizer and $h$ is a convex regularizer. Both $g$ and $h$ can be possibly nonsmooth. To solve the above regularized minimax problem, we study a proximal-GDA algorithm that leverages the forward-backward splitting update Lions and Mercier (1979); Attouch et al. (2013).

We study the variable convergence property of proximal-GDA in solving the minimax problem (P). Specifically, we show that proximal-GDA admits a novel Lyapunov function $H(x, y)$ (see Proposition 2), which is monotonically decreasing along the trajectory of proximal GDA, i.e., $H(x_{t+1}, y_{t+1}) < H(x_t, y_t)$. Based on the monotonicity of this Lyapunov function, we show that every limit point of the variable sequences generated by proximal-GDA is a critical point of the objective function.

Moreover, by exploiting the ubiquitous KŁ geometry of the Lyapunov function, we prove that the entire variable sequence of proximal-GDA has a unique limit point, or equivalently speaking, it

converges to a certain critical point $x^*$, i.e., $x_t \to x^*$, $y_t \to y^*(x^*)$ (see the definition of $y^*$ in Section 2). To the best of our knowledge, this is the first variable convergence result of GDA-type algorithms in nonconvex minimax optimization.

Furthermore, we characterize the asymptotic convergence rates of both the variable sequences and the function values of proximal-GDA in different parameterization regimes of the KŁ geometry. Depending on the value of the KŁ parameter $\theta$, we show that proximal-GDA achieves different types of convergence rates ranging from sublinear convergence up to finite-step convergence, as we summarize in Table 1 below.

Table 1: Convergence rates of proximal-GDA under different parameterizations of KŁ geometry. Note that $t_0$ denotes a sufficiently large positive integer.

| KŁ parameter | Function value convergence rate | Variable convergence rate |
|:---:|:---:|:---:|
| $\theta = 1$ | Finite-step convergence | Finite-step convergence |
| $\theta \in (\frac{1}{2}, 1)$ | $\mathcal{O}\big( \exp\big( - [2(1-\theta)]^{t_0-t} \big) \big)$ 
 Super-linear convergence | $\mathcal{O}\big( \exp\big( - [2(1-\theta)]^{t_0-t} \big) \big)$ 
 Super-linear convergence |
| $\theta = \frac{1}{2}$ | $\mathcal{O}\big( (1+\rho)^{t_0-t} \big), \rho > 0$ 
 Linear convergence | $\mathcal{O}\big( \big( \min\{2, 1+\rho\} \big)^{(t_0-t)/2} \big), \rho > 0$ 
 Linear convergence |
| $\theta \in (0, \frac{1}{2})$ | $\mathcal{O}\big( (t - t_0)^{-\frac{1}{1-2\theta}} \big)$ 
 Sub-linear convergence | $\mathcal{O}\big( (t - t_0)^{-\frac{\theta}{1-2\theta}} \big)$ 
 Sub-linear convergence |

## 1.2 RELATED WORK

**Deterministic GDA algorithms:** Yang et al. (2020) studied an alternating gradient descent-ascent (AGDA) algorithm in which the gradient ascent step uses the current variable $x_{t+1}$ instead of $x_t$. Boţ and Böhm (2020) extended the AGDA algorithm to an alternating proximal-GDA (APGDA) algorithm for a regularized minimax optimization. Xu et al. (2020b) studied an alternating gradient projection algorithm which applies $\ell_2$ regularizer to the local objective function of GDA followed by projection onto the constraint sets. Daskalakis and Panageas (2018); Mokhtari et al. (2020); Zhang and Wang (2020) analyzed optimistic gradient descent-ascent (OGDA) which applies negative momentum to accelerate GDA. Mokhtari et al. (2020) also studied an extra-gradient algorithm which applies two-step GDA in each iteration. Nouiehed et al. (2019) studied multi-step GDA where multiple gradient ascent steps are performed, and they also studied the momentum-accelerated version. Cherukuri et al. (2017); Daskalakis and Panageas (2018); Jin et al. (2020) studied GDA in continuous time dynamics using differential equations. Adolphs et al. (2019) analyzed a second-order variant of the GDA algorithm.

**Stochastic GDA algorithms:** Lin et al. (2020); Yang et al. (2020); Boţ and Böhm (2020) analyzed stochastic GDA, stochastic AGDA and stochastic APGDA, which are direct extensions of GDA, AGDA and APGDA to the stochastic setting respectively. Variance reduction techniques have been applied to stochastic minimax optimization, including SVRG-based Du and Hu (2019); Yang et al. (2020), SPIDER-based Xu et al. (2020a), STORM Qiu et al. (2020) and its gradient free version Huang et al. (2020). Xie et al. (2020) studied the complexity lower bound of first-order stochastic algorithms for finite-sum minimax problem.

**KŁ geometry:** The KŁ geometry was defined in Bolte et al. (2007). The KŁ geometry has been exploited to study the convergence of various first-order algorithms for solving minimization problems, including gradient descent Attouch and Bolte (2009), alternating gradient descent Bolte et al. (2014), distributed gradient descent Zhou et al. (2016a; 2018a), accelerated gradient descent Li et al. (2017). It has also been exploited to study the convergence of second-order algorithms such as Newton's method Noll and Rondepierre (2013); Frankel et al. (2015) and cubic regularization method Zhou et al. (2018b).

## 2 PROBLEM FORMULATION AND KŁ GEOMETRY

In this section, we introduce the problem formulation, technical assumptions and the Kurdyka-Łojasiewicz (KŁ) geometry. We consider the following regularized minimax optimization problem.

$$\min_{x \in \mathbb{R}^m} \max_{y \in \mathcal{Y}} \ f(x, y) + g(x) - h(y), \tag{P}$$

where $f : \mathbb{R}^m \times \mathbb{R}^n \to \mathbb{R}$ is a differentiable and nonconvex-strongly-concave loss function, $\mathcal{Y} \subset \mathbb{R}^n$ is a compact and convex set, and $g, h$ are the regularizers that are possibly non-smooth. In particular, define $\Phi(x) := \max_{y \in \mathcal{Y}} f(x, y) - h(y)$, and then the problem (P) is equivalent to the minimization problem $\min_{x \in \mathbb{R}^m} \Phi(x) + g(x)$.

Throughout the paper, we adopt the following standard assumptions on the problem (P).

**Assumption 1.** *The objective function of the problem* (P) *satisfies:*

1. *Function $f(\cdot, \cdot)$ is $L$-smooth and function $f(x, \cdot)$ is $\mu$-strongly concave;*
2. *Function $(\Phi + g)(x)$ is bounded below, i.e., $\inf_{\mathbf{x} \in \mathbb{R}^m}(\Phi + g)(x) > -\infty$;*
3. *For any $\alpha \in \mathbb{R}$, the sub-level set $\{x : (\Phi + g)(x) \le \alpha\}$ is compact;*
4. *Function $h$ is proper and convex, and function $g$ is proper and lower semi-continuous.*

To elaborate, item 1 considers the class of nonconvex-strongly-concave functions $f$ that has been widely studied in the existing literature Lin et al. (2020); Jin et al. (2020); Xu et al. (2020b;a); Lu et al. (2020). Items 2 and 3 guarantee that the minimax problem (P) has at least one solution, and the variable sequences generated by the proximal-GDA algorithm (See Algorithm 1) are bounded. Item 4 requires the regularizer $h$ to be convex (possibly non-smooth), which includes many norm-based popular regularizers such as $\ell_p$ $(p \ge 1)$, elastic net, nuclear norm, spectral norm, etc. On the other hand, the other regularizer $g$ can be nonconvex but lower semi-continuous, which includes all the aforementioned convex regularizers, $\ell_p$ $(0 \le p < 1)$, Schatten-$p$ norm, rank, etc. Hence, our formulation of the problem (P) covers a rich class of nonconvex objective functions and regularizers and is more general than the existing nonconvex minimax formulation in Lin et al. (2020), which does not consider any regularizer.

**Remark 1.** *We note that the strong concavity of $f(x, \cdot)$ in item 1 can be relaxed to concavity, provided that the regularizer $h(y)$ is $\mu$-strongly convex. In this case, we can add $-\frac{\mu}{2}\|y\|^2$ to both $f(x, y)$ and $h(y)$ such that Assumption 1 still holds. For simplicity, we will omit the discussion on this case.*

By strong concavity of $f(x, \cdot)$, it is clear that the mapping $y^*(x) := \arg\max_{y \in \mathcal{Y}} f(x, y) - h(y)$ is uniquely defined for every $x \in \mathbb{R}^m$. In particular, if $x^*$ is the desired minimizer of $\Phi(x)$, then $(x^*, y^*(x^*))$ is the desired solution of the minimax problem (P).

Next, we present some important properties regarding the function $\Phi(x)$ and the mapping $y^*(x)$. The following proposition from Boţ and Böhm (2020) generalizes the Lemma 4.3 of Lin et al. (2020) to the regularized setting. The proof can be found in Appendix A. Throughout, we denote $\kappa = L/\mu$ as the condition number and denote $\nabla_1 f(x, y), \nabla_2 f(x, y)$ as the gradients with respect to the first and the second input argument, respectively. For example, with this notation, $\nabla_1 f(x, y^*(x))$ denotes the gradient of $f(x, y^*(x))$ with respect to only the first input argument $x$, and the $x$ in the second input argument $y^*(x)$ is treated as a constant.

**Proposition 1** (Lipschitz continuity of $y^*(x)$ and $\nabla\Phi(x)$). *Let Assumption 1 hold. Then, the mapping $y^*(x)$ and the function $\Phi(x)$ satisfy*

1. *Mapping $y^*(x)$ is $\kappa$-Lipschitz continuous;*
2. *Function $\Phi(x)$ is $L(1 + \kappa)$-smooth with $\nabla\Phi(x) = \nabla_1 f(x, y^*(x))$.*

As an intuitive explanation of Proposition 1, since the function $f(x, y) - h(y)$ is $L$-smooth with respect to $x$, both the maximizer $y^*(x)$ and the corresponding maximum function value $\Phi(x)$ should not change substantially with regard to a small change of $x$.

Recall that the minimax problem (P) is equivalent to the standard minimization problem $\min_{x \in \mathbb{R}^m} \Phi(x) + g(x)$, which, according to item 2 of Proposition 1, includes a smooth nonconvex function $\Phi(x)$ and a lower semi-continuous regularizer $g(x)$. Hence, we can define the optimization goal of the minimax problem (P) as **finding a critical point $x^*$ of the nonconvex function** $\Phi(x) + g(x)$ that satisfies the necessary optimality condition $\mathbf{0} \in \partial(\Phi + g)(x^*)$ for minimizing nonconvex functions. Here, $\partial$ denotes the notion of subdifferential as we elaborate below.

**Definition 1.** *(Subdifferential and critical point, Rockafellar and Wets (2009)) The Frechét subdifferential $\widehat{\partial}h$ of function $h$ at $x \in \mathrm{dom}\, h$ is the set of $u \in \mathbb{R}^d$ defined as*

$$\widehat{\partial}h(x) := \Big\{ u : \liminf_{z \neq x, z \to x} \frac{h(z) - h(x) - u^{\mathsf{T}}(z-x)}{\|z - x\|} \geq 0 \Big\},$$

*and the limiting subdifferential $\partial h$ at $x \in \mathrm{dom}\, h$ is the graphical closure of $\widehat{\partial}h$ defined as:*

$$\partial h(x) := \{ u : \exists x_k \to x, h(x_k) \to h(x), u_k \in \widehat{\partial}h(x_k), u_k \to u \}.$$

*The set of **critical points** of $h$ is defined as $\mathrm{crit}\, h := \{ x : \mathbf{0} \in \partial h(x) \}$.*

Throughout, we refer to the limiting subdifferential as subdifferential. We note that subdifferential is a generalization of gradient (when $h$ is differentiable) and subgradient (when $h$ is convex) to the nonconvex setting. In particular, any local minimizer of $h$ must be a critical point.

Next, we introduce the Kurdyka-Łojasiewicz (KŁ) geometry of a function $h$. Throughout, the point-to-set distance is denoted as $\mathrm{dist}_\Omega(x) := \inf_{u \in \Omega} \|x - u\|$.

**Definition 2** (KŁ geometry, Bolte et al. (2014))**.** *A proper and lower semi-continuous function $h$ is said to have the KŁ geometry if for every compact set $\Omega \subset \mathrm{dom}\, h$ on which $h$ takes a constant value $h_\Omega \in \mathbb{R}$, there exist $\varepsilon, \lambda > 0$ such that for all $\bar{x} \in \Omega$ and all $x \in \{ z \in \mathbb{R}^m : \mathrm{dist}_\Omega(z) < \varepsilon, h_\Omega < h(z) < h_\Omega + \lambda \}$, the following condition holds:*

$$\varphi'\left(h(x) - h_\Omega\right) \cdot \mathrm{dist}_{\partial h(x)}(\mathbf{0}) \geq 1, \tag{1}$$

*where $\varphi'$ is the derivative of function $\varphi : [0, \lambda) \to \mathbb{R}_+$, which takes the form $\varphi(t) = \frac{c}{\theta} t^\theta$ for certain universal constant $c > 0$ and KŁ parameter $\theta \in (0, 1]$.*

The KŁ geometry characterizes the local geometry of a nonconvex function around the set of critical points. To explain, consider the case where $h$ is a differentiable function so that $\partial h(x) = \nabla h(x)$. Then, the KŁ inequality in eq. (1) becomes $h(x) - h_\Omega \leq \mathcal{O}(\|\nabla h(x)\|^{\frac{1}{1-\theta}})$, which generalizes the Polyak-Łojasiewicz (PL) condition $h(x) - h_\Omega \leq \mathcal{O}(\|\nabla h(x)\|^2)$ Łojasiewicz (1963); Karimi et al. (2016) (i.e., KŁ parameter $\theta = \frac{1}{2}$). Moreover, the KŁ geometry has been shown to hold for a large class of functions including sub-analytic functions, logarithm and exponential functions and semi-algebraic functions. These function classes cover most of the nonconvex objective functions encountered in practical machine learning applications Zhou et al. (2016b); Yue et al. (2018); Zhou and Liang (2017); Zhou et al. (2018b).

The KŁ geometry has been exploited extensively to analyze the convergence of various first-order algorithms, e.g., gradient descent Attouch and Bolte (2009); Li et al. (2017), alternating minimization Bolte et al. (2014) and distributed gradient methods Zhou et al. (2016a). It has also been exploited to study the convergence of second-order algorithms such cubic regularization Zhou et al. (2018b). In these works, it has been shown that the variable sequences generated by these algorithms converge to a desired critical point in nonconvex optimization, and the convergence rates critically depend on the parameterization $\theta$ of the KŁ geometry. In the subsequent sections, we provide a comprehensive understanding of the convergence and convergence rate of proximal-GDA under the KŁ geometry.

## 3 PROXIMAL-GDA AND GLOBAL CONVERGENCE ANALYSIS

In this section, we study the following proximal-GDA algorithm that leverages the forward-backward splitting updates Lions and Mercier (1979); Attouch et al. (2013) to solve the regularized minimax problem (P) and analyze its global convergence properties. In particular, the proximal-GDA algorithm is a generalization of the GDA Du and Hu (2019) and projected GDA Nedić and Ozdaglar (2009) algorithms. The algorithm update rule is specified in Algorithm 1, where the two proximal gradient steps are formally defined as

$$\mathrm{prox}_{\eta_x g}\big(x_t - \eta_x \nabla_1 f(x_t, y_t)\big) :\in \operatorname*{argmin}_{u \in \mathbb{R}^m} \Big\{ g(u) + \frac{1}{2\eta_x} \|u - x_t + \eta_x \nabla_1 f(x_t, y_t)\|^2 \Big\}, \tag{2}$$

$$\mathrm{prox}_{\eta_y h}\big(y_t + \eta_y \nabla_2 f(x_t, y_t)\big) := \operatorname*{argmin}_{v \in \mathcal{Y}} \Big\{ h(v) + \frac{1}{2\eta_y} \|v - y_t - \eta_y \nabla_2 f(x_t, y_t)\|^2 \Big\}, \tag{3}$$

---

**Algorithm 1** Proximal-GDA

---

**Input:** Initialization $x_0, y_0$, learning rates $\eta_x, \eta_y$.
**for** $t = 0, 1, 2, \ldots, T - 1$ **do**

$$x_{t+1} \in \text{prox}_{\eta_x g}\big(x_t - \eta_x \nabla_1 f(x_t, y_t)\big),$$
$$y_{t+1} = \text{prox}_{\eta_y h}\big(y_t + \eta_y \nabla_2 f(x_t, y_t)\big).$$

**end**
**Output:** $x_T, y_T$.

---

Recall that our goal is to obtain a critical point of the minimization problem $\min_{x \in \mathbb{R}^m} \Phi(x) + g(x)$. Unlike the gradient descent algorithm which generates a sequence of monotonically decreasing function values, the function value $(\Phi + g)(x_k)$ along the variable sequence generated by proximal-GDA is generally oscillating due to the alternation between the gradient descent and gradient ascent steps. Hence, it seems that proximal-GDA is less stable than gradient descent. However, our next result shows that, for the problem (P), the proximal-GDA admits a special Lyapunov function that monotonically decreases in the optimization process. The proof of Proposition 2 is in Appendix B.

**Proposition 2.** *Let Assumption 1 hold and define the Lyapunov function* $H(z) := \Phi(x) + g(x) + (1 - \frac{1}{4\kappa^2})\|y - y^*(x)\|^2$ *with* $z := (x, y)$. *Choose the learning rates such that* $\eta_x \leq \frac{1}{\kappa^3(L+3)^2}$, $\eta_y \leq \frac{1}{L}$. *Then, the variables* $z_t = (x_t, y_t)$ *generated by proximal-GDA satisfy, for all* $t = 0, 1, 2, \ldots$

$$H(z_{t+1}) \leq H(z_t) - 2\|x_{t+1} - x_t\|^2 - \frac{1}{4\kappa^2}\big(\|y_{t+1} - y^*(x_{t+1})\|^2 + \|y_t - y^*(x_t)\|^2\big). \quad (4)$$

We first explain how this Lyapunov function is introduced in the proof. By eq. (19) in the supplementary material, we established a recursive inequality on the objective function $(\Phi + g)(x_{t+1})$. One can see that the right hand side of eq. (19) contains a negative term $-\|x_{t+1} - x_t\|^2$ and an undesired positive term $\|y^*(x_t) - y_t\|^2$. Hence, the objective function $(\Phi + g)(x_{t+1})$ may be oscillating and cannot serve as a proper Lyapunov function. In the subsequent analysis, we break this positive term into a difference of two terms $\|y^*(x_t) - y_t\|^2 - \|y^*(x_{t+1}) - y_{t+1}\|^2$, by leveraging the update of $y_{t+1}$ for solving the strongly concave maximization problem. After proper rearranging, this difference term contributes to the quadratic term in the Lyapunov function.

We note that the Lyapunov function $H(z)$ is the objective function $\Phi(x) + g(x)$ regularized by the additional quadratic term $(1 - \frac{1}{4\kappa^2})\|y - y^*(x)\|^2$, and such a Lyapunov function clearly characterizes our optimization goal. To elaborate, consider a desired case where the sequence $x_t$ converges to a certain critical point $x^*$ and the sequence $y_t$ converges to the corresponding point $y^*(x^*)$. In this case, it can be seen that the Lyapunov function $H(z_t)$ converges to the desired function value $(\Phi + g)(x^*)$. Hence, solving the minimax problem (P) is equivalent to minimizing the Lyapunov function. More importantly, Proposition 2 shows that the Lyapunov function value sequence $\{H(z_t)\}_t$ is monotonically decreasing in the optimization process of proximal-GDA, implying that the algorithm continuously makes optimization progress. We also note that the coefficient $(1 - \frac{1}{4\kappa^2})$ in the Lyapunov function is chosen in a way so that eq. (4) can be proven to be strictly decreasing. This monotonic property is the core of our analysis of proximal-GDA.

Based on Proposition 2, we obtain the following asymptotic properties of the variable sequences generated by proximal-GDA. The proof can be found in Appendix C.

**Corollary 1.** *Based on Proposition 2, the sequences* $\{x_t, y_t\}_t$ *generated by proximal-GDA satisfy*

$$\lim_{t \to \infty} \|x_{t+1} - x_t\| = 0, \quad \lim_{t \to \infty} \|y_{t+1} - y_t\| = 0, \quad \lim_{t \to \infty} \|y_t - y^*(x_t)\| = 0.$$

The above result shows that the variable sequences generated by proximal-GDA in solving the problem (P) are asymptotically stable. In particular, the last two equations show that $y_t$ asymptotically approaches the corresponding maximizer $y^*(x_t)$ of the objective function $f(x_t, y) + g(x_t) - h(y)$. Hence, if $x_t$ converges to a certain critical point, $y_t$ will converge to the corresponding maximizer.

**Discussion:** We note that the monotonicity property in Proposition 2 further implies the convergence rate result $\min_{0 \leq k \leq t} \|x_{k+1} - x_k\| \leq \mathcal{O}(t^{-1/2})$ (by telescoping over $t$). When there is no regularizer,

this convergence rate result can be shown to further imply that $\min_{0 \le k \le t} \|\nabla \Phi(x_k)\| \le \mathcal{O}(t^{-1/2})$, which reduces to the Theorem 4.4 of Lin et al. (2020). However, such a convergence rate result does not imply the convergence of the variable sequences $\{x_t\}_t, \{y_t\}_t$. To explain, we can apply the convergence rate result $\|x_{t+1} - x_t\| \le \mathcal{O}(t^{-1/2})$ to bound the trajectory norm as $\|x_T\| \le \|x_0\| + \sum_{t=0}^{T-1} \|x_{t+1} - x_t\| \approx \sqrt{T}$, which diverges to $+\infty$ as $T \to \infty$. Therefore, such a type of convergence rate does not even imply the boundedness of the trajectory. In this paper, our focus is to establish the convergence of the variable sequences generated by proximal-GDA.

All the results in Corollary 1 imply that the alternating proximal gradient descent & ascent updates of proximal-GDA can achieve stationary points, which we show below to be critical points.

**Theorem 1** (Global convergence). *Let Assumption 1 hold and choose the learning rates $\eta_x \le \frac{1}{\kappa^3(L+3)^2}$, $\eta_y \le \frac{1}{L}$. Then, proximal-GDA satisfies the following properties.*

1. *The function value sequence $\{(\Phi + g)(x_t)\}_t$ converges to a finite limit $H^* > -\infty$;*
2. *The sequences $\{x_t\}_t, \{y_t\}_t$ are bounded and have compact sets of limit points. Moreover, $(\Phi + g)(x^*) \equiv H^*$ for any limit point $x^*$ of $\{x_t\}_t$;*
3. *Every limit point of $\{x_t\}_t$ is a critical point of $(\Phi + g)(x)$.*

The proof of Theorem 1 is presented in Appendix D. The above theorem establishes the global convergence property of proximal-GDA. Specifically, item 1 shows that the function value sequence $\{(\Phi + g)(x_t)\}_t$ converges to a finite limit $H^*$, which is also the limit of the Lyapunov function sequence $\{H(z_t)\}_t$. Moreover, items 2 & 3 further show that all the converging subsequences of $\{x_t\}_t$ converge to critical points of the problem, at which the function $\Phi + g$ achieves the constant value $H^*$. These results show that proximal-GDA can properly find critical points of the minimax problem (P). Furthermore, based on these results, the variable sequences generated by proximal-GDA are guaranteed to enter a local parameter region where the Kurdyka-Łojasiewicz geometry holds, which we exploit in the next section to establish stronger convergence results of the algorithm.

## 4 Variable Convergence of Proximal-GDA under KŁ Geometry

We note that Theorem 1 only shows that every limit point of $\{x_t\}_t$ is a critical point, and the sequences $\{x_t, y_t\}_t$ may not necessarily be convergent. In this section, we exploit the local KŁ geometry of the Lyapunov function to formally prove the convergence of these sequences. Throughout this section, we adopt the following assumption.

**Assumption 2.** *Regarding the mapping $y^*(x)$, the function $\|y^*(x) - y\|^2$ has a non-empty subdifferential, i.e., $\partial_x(\|y^*(x) - y\|^2) \ne \emptyset$.*

Note that in many practical scenarios $y^*(x)$ is sub-differentiable. In addition, Assumption 2 ensures the sub-differentiability of the Lyapunov function $H(z) := \Phi(x) + g(x) + (1 - \frac{1}{4\kappa^2})\|y - y^*(x)\|^2$. We obtain the following variable convergence result of proximal-GDA under the KŁ geometry. The proof is presented in Appendix E.

**Theorem 2** (Variable convergence). *Let Assumption 1 & 2 hold and assume that $H$ has the KŁ geometry. Choose the learning rates $\eta_x \le \frac{1}{\kappa^3(L+3)^2}$ and $\eta_y \le \frac{1}{L}$. Then, the sequence $\{(x_t, y_t)\}_t$ generated by proximal-GDA converges to a certain critical point $(x^*, y^*(x^*))$ of $(\Phi + g)(x)$, i.e.,*

$$x_t \xrightarrow{t} x^*, \quad y_t \xrightarrow{t} y^*(x^*).$$

Theorem 2 formally shows that proximal-GDA is guaranteed to converge to a certain critical point $(x^*, y^*(x^*))$ of the minimax problem (P), provided that the Lyapunov function belongs to the large class of KŁ functions. To the best of our knowledge, this is the first variable convergence result of GDA-type algorithms in nonconvex minimax optimization. The proof logic of Theorem 2 can be summarized as the following two key steps.

**Step 1:** By leveraging the monotonicity property of the Lyapunov function in Proposition 2, we first show that the variable sequences of proximal-GDA eventually enter a local region where the KŁ geometry holds;

**Step 2:** Then, combining the KŁ inequality in eq. (1) and the monotonicity property of the Lyapunov function in eq. (4), we show that the variable sequences of proximal-GDA are Cauchy sequences and hence converge to a certain critical point.

## 5 CONVERGENCE RATE OF PROXIMAL-GDA UNDER KŁ GEOMETRY

In this section, we exploit the parameterization of the KŁ geometry to establish various types of asymptotic convergence rates of proximal-GDA.

We obtain the following asymptotic convergence rates of proximal-GDA under different parameter regimes of the KŁ geometry. The proof is presented in Appendix F. In the sequel, we denote $t_0$ as a sufficiently large positive integer, denote $c > 0$ as the constant in Definition 2 and also define

$$M := \max\left\{\frac{1}{2}\left(\frac{1}{\eta_x} + (L + 4\kappa^2)(1 + \kappa)\right)^2, 4\kappa^2(L + 4\kappa)^2\right\}. \tag{5}$$

**Theorem 3** (Funtion value convergence rate). *Under the same conditions as those of Theorem 2, the Lyapunov function value sequence $\{H(z_t)\}_t$ converges to the limit $H^*$ at the following rates.*

*1. If KŁ geometry holds with $\theta = 1$, then $H(z_t) \downarrow H^*$ within **finite number of iterations**;*
*2. If KŁ geometry holds with $\theta \in (\frac{1}{2}, 1)$, then $H(z_t) \downarrow H^*$ **super-linearly** as*

$$H(z_t) - H^* \leq (2Mc^2)^{-\frac{1}{2\theta-1}} \exp\left(-\left(\frac{1}{2(1-\theta)}\right)^{t-t_0}\right), \quad \forall t \geq t_0; \tag{6}$$

*3. If KŁ geometry holds with $\theta = \frac{1}{2}$, then $H(z_t) \downarrow H^*$ **linearly** as*

$$H(z_t) - H^* \leq \left(1 + \frac{1}{2Mc^2}\right)^{t_0-t}(H(z_{t_0}) - H^*), \quad \forall t \geq t_0; \tag{7}$$

*4. If KŁ geometry holds with $\theta \in (0, \frac{1}{2})$, then $H(z_t) \downarrow H^*$ **sub-linearly** as*

$$H(z_t) - H^* \leq \left[C(t - t_0)\right]^{-\frac{1}{1-2\theta}}, \quad \forall t \geq t_0. \tag{8}$$

*where $C = \min\left[\frac{1-2\theta}{8Mc^2}, d_{t_0}^{-(1-2\theta)}\left(1 - 2^{-(1-2\theta)}\right)\right] > 0$.*

It can be seen from the above theorem that the convergence rate of the Lyapunov function of proximal-GDA is determined by the KŁ parameter $\theta$. A larger $\theta$ implies that the local geometry of $H$ is 'sharper', and hence the corresponding convergence rate is orderwise faster. In particular, the algorithm converges at a linear rate when the KŁ geometry holds with $\theta = \frac{1}{2}$ (see the item 3), which is a generalization of the Polyak-Łojasiewicz (PL) geometry. As a comparison, in the existing analysis of GDA, such a linear convergence result is established under stronger geometries, e.g., convex-strongly-concave Du and Hu (2019), strongly-convex-strongly-concave Mokhtari et al. (2020); Zhang and Wang (2020) and two-sided PL condition Yang et al. (2020). In summary, the above theorem provides a full characterization of the fast convergence rates of proximal-GDA in the full spectrum of the KŁ geometry.

Moreover, we also obtain the following asymptotic convergence rates of the variable sequences that are generated by proximal-GDA under different parameterization of the KŁ geometry. The proof is presented in Appendix G.

**Theorem 4** (Variable convergence rate). *Under the same conditions as those of Theorem 2, the sequences $\{x_t, y_t\}_t$ converge to their limits $x^*, y^*(x^*)$ respectively at the following rates.*

*1. If KŁ geometry holds with $\theta = 1$, then $(x_t, y_t) \to (x^*, y^*(x^*))$ within **finite number of iterations**;*
*2. If KŁ geometry holds with $\theta \in (\frac{1}{2}, 1)$, then $(x_t, y_t) \to (x^*, y^*(x^*))$ **super-linearly** as*

$$\max\left\{\|x_t - x^*\|, \|y_t - y^*(x^*)\|\right\} \leq \mathcal{O}\left(\exp\left(-\left(\frac{1}{2(1-\theta)}\right)^{t-t_0}\right)\right), \quad \forall t \geq t_0; \tag{9}$$

*3. If KŁ geometry holds with $\theta = \frac{1}{2}$, then $(x_t, y_t) \to (x^*, y^*(x^*))$ **linearly** as*

$$\max\left\{\|x_t - x^*\|, \|y_t - y^*(x^*)\|\right\} \leq \mathcal{O}\left(\left(\min\left\{2, 1 + \frac{1}{2Mc^2}\right\}\right)^{(t_0-t)/2}\right), \quad \forall t \geq t_0; \tag{10}$$

*4. If KŁ geometry holds with $\theta \in (0, \frac{1}{2})$, then $(x_t, y_t) \to (x^*, y^*(x^*))$ **sub-linearly** as*

$$\max\left\{\|x_t - x^*\|, \|y_t - y^*(x^*)\|\right\} \leq \mathcal{O}\left((t - t_0)^{-\frac{\theta}{1-2\theta}}\right), \quad \forall t \geq t_0. \tag{11}$$

To the best of our knowledge, this is the first characterization of the variable convergence rates of proximal-GDA in the full spectrum of the KŁ geometry. It can be seen that, similar to the convergence rate results of the function value sequence, the convergence rate of the variable sequences is also affected by the parameterization of the KŁ geometry.

## 6 CONCLUSION

In this paper, we develop a new analysis framework for the proximal-GDA algorithm in nonconvex-strongly-concave optimization. Our key observation is that proximal-GDA has a intrinsic Lyapunov function that monotonically decreases in the minimax optimization process. Such a property demonstrates the stability of the algorithm. Moreover, we establish the formal variable convergence of proximal-GDA to a critical point of the objective function under the ubiquitous KŁ geometry. Our results fully characterize the impact of the parameterization of the KŁ geometry on the convergence rate of the algorithm. In the future study, we will leverage such an analysis framework to explore the convergence of stochastic GDA algorithms and their variance-reduced variants.

## ACKNOWLEDGEMENT

The work of T. Xu and Y. Liang was supported partially by the U.S. National Science Foundation under the grants CCF-1900145 and CCF-1909291.

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

SUPPLEMENTARY MATERIAL

## A PROOF OF PROPOSITION 1

**Proposition 1** (Lipschitz continuity of $y^*(x)$ and $\nabla\Phi(x)$). *Let Assumption 1 hold. Then, the mapping $y^*(x)$ and the function $\Phi(x)$ satisfy*

*1. Mapping $y^*(x)$ is $\kappa$-Lipschitz continuous;*
*2. Function $\Phi(x)$ is $L(1+\kappa)$-smooth with $\nabla\Phi(x) = \nabla_1 f(x, y^*(x))$.*

*Proof.* We first prove item 1. Since $f(x, y)$ is strongly concave in $y$ for every $x$ and $h(y)$ is convex, the mapping $y^*(x) = \arg\max_{y \in \mathcal{Y}} f(x, y) - h(y)$ is uniquely defined. We first show that $y^*(x)$ is a Lipschitz mapping. Consider two arbitrary points $x_1, x_2$. The optimality conditions of $y^*(x_1)$ and $y^*(x_2)$ imply that

$$\langle y - y^*(x_1), \nabla_2 f(x_1, y^*(x_1)) - u_1 \rangle \leq 0, \quad \forall y \in \mathcal{Y}, u_1 \in \partial h(y^*(x_1)), \tag{12}$$

$$\langle y - y^*(x_2), \nabla_2 f(x_2, y^*(x_2)) - u_2 \rangle \leq 0, \quad \forall y \in \mathcal{Y}, u_2 \in \partial h(y^*(x_2)). \tag{13}$$

Setting $y = y^*(x_2)$ in eq. (12), $y = y^*(x_1)$ in eq. (13) and summing up the two inequalities, we obtain that

$$\langle y^*(x_2) - y^*(x_1), \nabla_2 f(x_1, y^*(x_1)) - \nabla_2 f(x_2, y^*(x_2)) - u_1 + u_2 \rangle \leq 0. \tag{14}$$

Since $\partial h$ is a monotone operator (by convexity), we know that $\langle u_2 - u_1, y^*(x_2) - y^*(x_1) \rangle \geq 0$. Hence, the above inequality further implies that

$$\langle y^*(x_2) - y^*(x_1), \nabla_2 f(x_1, y^*(x_1)) - \nabla_2 f(x_2, y^*(x_2)) \rangle \leq 0. \tag{15}$$

Next, by strong concavity of $f(x_1, \cdot)$, we have that

$$\langle y^*(x_2) - y^*(x_1), \nabla_2 f(x_1, y^*(x_2)) - \nabla_2 f(x_1, y^*(x_1)) \rangle + \mu\|y^*(x_1) - y^*(x_2)\|^2 \leq 0. \tag{16}$$

Adding up the above two inequalities yields that

$$\begin{aligned}
\mu\|y^*(x_1) - y^*(x_2)\|^2 &\leq \langle y^*(x_2) - y^*(x_1), \nabla_2 f(x_2, y^*(x_2)) - \nabla_2 f(x_1, y^*(x_2)) \rangle \\
&\leq \|y^*(x_2) - y^*(x_1)\| \|\nabla_2 f(x_2, y^*(x_2)) - \nabla_2 f(x_1, y^*(x_2))\| \\
&\leq L\|y^*(x_2) - y^*(x_1)\| \|x_2 - x_1\|.
\end{aligned}$$

The above inequality shows that $\|y^*(x_1) - y^*(x_2)\| \leq \kappa\|x_2 - x_1\|$, and item 1 is proved.

Next, we will prove item 2.

Consider $A_n = \{y^*(x) : x \in \mathbb{R}^m, \|x\| \leq n\} \subset \mathcal{Y}$. Since $h$ is proper and convex, $h(y_0) < +\infty$ for some $y_0 \in \mathcal{Y}$. Since $f$ is $L$-smooth, its value is finite everywhere. Hence, for any $x \in \mathbb{R}^m$, $\Phi(x) = \max_{y \in \mathcal{Y}} f(x, y) - h(y) \geq f(x, y_0) - h(y_0) > -\infty$, so $h(y^*(x)) = f(x, y^*(x)) - \Phi(x) < +\infty$. Therefore, based on Corollary 10.1.1 of Rockafellar (1970), $h(y)$ is continuous on $A_n$ and thus $f(x, y) - h(y)$ is continuous in $(x, y) \in \mathbb{R}^m \times A_n$. Also, $\nabla_1 f(x, y)$ is continuous in $(x, y) \in \mathbb{R}^m \times A_n$ since $f$ is $L$-smooth. For any sequence $\{x_k\}$ such that $\|x_k\| \leq n$ and $y^*(x_k) \to y \in \mathcal{Y}$, $y = y^*(x')$ for any limit point $x'$ of $\{x_k\}$ (there is at least one such limit point since $\|x_k\| \leq n$) since we have proved that $y^*$ is continuous. As $\|x'\| \leq n$, $y \in A_n$. Hence, $A_n$ is closed. As $A_n$ is included in bounded $\mathcal{Y}$, $A_n$ is compact. Therefore, based on the Danskin theorem Bernhard and Rapaport (1995), the function $\Phi_n(x) := \arg\max_{y \in A_n} f(x, y) - h(y)$ is differntialable with $\nabla\Phi_n(x) = \nabla_1 f(x, y^*(x))$. On one hand, $\Phi_n(x) \leq \Phi(x)$ since $A_n \subset \mathcal{Y}$. On the other hand, when $\|x\| \leq n$, $y^*(x) \in A_n$, so $\Phi(x) = f(x, y^*(x)) - h(y^*(x)) \leq \Phi_n(x)$. Hence, when $\|x\| \leq n$, $\Phi(x) = \Phi_n(x)$ and thus $\nabla\Phi(x) = \nabla\Phi_n(x) = \nabla_1 f(x, y^*(x))$. Since $n$ can be arbitrarily large, $\nabla\Phi(x) = \nabla_1 f(x, y^*(x))$ for any $x \in \mathbb{R}^m$.

Next, consider any $x_1, x_2 \in \mathbb{R}^m$, we obtain that

$$\begin{aligned}
\|\nabla\Phi(x_2) - \nabla\Phi(x_1)\| &= \|\nabla_1 f(x_2, y^*(x_2)) - \nabla_1 f(x_1, y^*(x_1))\| \\
&\leq L\|x_2 - x_1\| + L\|y^*(x_2) - y^*(x_1)\| \\
&\leq L\|x_2 - x_1\| + L\kappa\|x_2 - x_1\| \\
&= L(1+\kappa)\|x_2 - x_1\|,
\end{aligned}$$

which implies that $\Phi(x)$ is $L(1+\kappa)$-smooth.

$\square$

## B    PROOF OF PROPOSITION 2

**Proposition 2.** *Let Assumption 1 hold and define the Lyapunov function $H(z) := \Phi(x) + g(x) + (1 - \frac{1}{4\kappa^2})\|y - y^*(x)\|^2$ with $z := (x, y)$. Choose the learning rates such that $\eta_x \le \frac{1}{\kappa^3(L+3)^2}$, $\eta_y \le \frac{1}{L}$. Then, the variables $z_t = (x_t, y_t)$ generated by proximal-GDA satisfy, for all $t = 0, 1, 2, ...$*

$$H(z_{t+1}) \le H(z_t) - 2\|x_{t+1} - x_t\|^2 - \frac{1}{4\kappa^2}\left(\|y_{t+1} - y^*(x_{t+1})\|^2 + \|y_t - y^*(x_t)\|^2\right). \quad (4)$$

*Proof.* Consider the $t$-th iteration of proximal-GDA. By smoothness of $\Phi$ we obtain that

$$\Phi(x_{t+1}) \le \Phi(x_t) + \langle x_{t+1} - x_t, \nabla\Phi(x_t)\rangle + \frac{L(1+\kappa)}{2}\|x_{t+1} - x_t\|^2. \quad (17)$$

On the other hand, by the definition of the proximal gradient step of $x_t$, we have

$$g(x_{t+1}) + \frac{1}{2\eta_x}\|x_{t+1} - x_t + \eta_x\nabla_1 f(x_t, y_t)\|^2 \le g(x_t) + \frac{1}{2\eta_x}\|\eta_x\nabla_1 f(x_t, y_t)\|^2,$$

which further simplifies to

$$g(x_{t+1}) \le g(x_t) - \frac{1}{2\eta_x}\|x_{t+1} - x_t\|^2 - \langle x_{t+1} - x_t, \nabla_1 f(x_t, y_t)\rangle. \quad (18)$$

Adding up eq. (17) and eq. (18) yields that

$\Phi(x_{t+1}) + g(x_{t+1})$

$$\le \Phi(x_t) + g(x_t) - \left(\frac{1}{2\eta_x} - \frac{L(1+\kappa)}{2}\right)\|x_{t+1} - x_t\|^2 + \langle x_{t+1} - x_t, \nabla\Phi(x_t) - \nabla_1 f(x_t, y_t)\rangle$$

$$= \Phi(x_t) + g(x_t) - \left(\frac{1}{2\eta_x} - \frac{L(1+\kappa)}{2}\right)\|x_{t+1} - x_t\|^2 + \|x_{t+1} - x_t\|\|\nabla\Phi(x_t) - \nabla_1 f(x_t, y_t)\|$$

$$= \Phi(x_t) + g(x_t) - \left(\frac{1}{2\eta_x} - \frac{L(1+\kappa)}{2}\right)\|x_{t+1} - x_t\|^2 + \|x_{t+1} - x_t\|\|\nabla_1 f(x_t, y^*(x_t)) - \nabla_1 f(x_t, y_t)\|$$

$$\le \Phi(x_t) + g(x_t) - \left(\frac{1}{2\eta_x} - \frac{L(1+\kappa)}{2}\right)\|x_{t+1} - x_t\|^2 + L\|x_{t+1} - x_t\|\|y^*(x_t) - y_t\|.$$

$$\le \Phi(x_t) + g(x_t) - \left(\frac{1}{2\eta_x} - \frac{L(1+\kappa)}{2} - \frac{L^2\kappa^2}{2}\right)\|x_{t+1} - x_t\|^2 + \frac{1}{2\kappa^2}\|y^*(x_t) - y_t\|^2 \quad (19)$$

Next, consider the term $\|y^*(x_t) - y_t\|$ in the above inequality. Note that $y^*(x_t)$ is the unique minimizer of the strongly concave function $f(x_t, y) - h(y)$, and $y_{t+1}$ is obtained by applying one proximal gradient step on it starting from $y_t$. Hence, by the convergence rate of proximal gradient ascent algorithm under strong concavity, we conclude that with $\eta_y \le \frac{1}{L}$,

$$\|y_{t+1} - y^*(x_t)\|^2 \le \left(1 - \kappa^{-1}\right)\|y_t - y^*(x_t)\|^2. \quad (20)$$

Hence, we further obtain that

$$\|y^*(x_{t+1}) - y_{t+1}\|^2 \le \left(1 + \kappa^{-1}\right)\|y_{t+1} - y^*(x_t)\|^2 + (1+\kappa)\|y^*(x_{t+1}) - y^*(x_t)\|^2$$
$$\le \left(1 - \kappa^{-2}\right)\|y_t - y^*(x_t)\|^2 + \kappa^2(1+\kappa)\|x_{t+1} - x_t\|^2. \quad (21)$$

Adding eqs. (19) & (21), we obtain

$$\Phi(x_{t+1}) + g(x_{t+1})$$
$$\le \Phi(x_t) + g(x_t) - \left(\frac{1}{2\eta_x} - \frac{L(1+\kappa)}{2} - \frac{L^2\kappa^2}{2} - \kappa^2(1+\kappa)\right)\|x_{t+1} - x_t\|^2$$
$$+ \left(1 - \frac{1}{2\kappa^2}\right)\|y^*(x_t) - y_t\|^2 - \|y^*(x_{t+1}) - y_{t+1}\|^2$$

Rearranging the equation above and recalling the definition of the Lyapunov function $H(z) := \Phi(x) + g(x) + \left(1 - \frac{1}{4\kappa^2}\right)\|y - y^*(x)\|^2$, we have

$$H(z_{t+1}) \leq H(z_t) - \left(\frac{1}{2\eta_x} - \frac{L(1+\kappa)}{2} - \frac{L^2\kappa^2}{2} - \kappa^2(1+\kappa)\right)\|x_{t+1} - x_t\|^2$$
$$- \frac{1}{4\kappa^2}(\|y^*(x_t) - y_t\|^2 + \|y^*(x_{t+1}) - y_{t+1}\|^2) \tag{22}$$

When $\eta_x < \kappa^{-3}(L+3)^{-2}$, using $\kappa \geq 1$ yields that

$$\frac{1}{2\eta_x} - \frac{L(1+\kappa)}{2} - \frac{L^2\kappa^2}{2} - \kappa^2(1+\kappa)$$
$$\geq \frac{1}{2}\kappa^3(L+3)^2 - \frac{L}{2}(2\kappa)\kappa^2 - \frac{L^2\kappa^3}{2} - \kappa^2(2\kappa)$$
$$= \frac{1}{2}\kappa^3[(L+3)^2 - 2L - L^2 - 4]$$
$$= \frac{1}{2}\kappa^3(4L+5) > 2 \tag{23}$$

As a result, eq. (4) can be concluded by substituting eq. (23) into eq. (22). $\qquad\square$

## C    PROOF OF COROLLARY 1

**Corollary 1.** *Based on Proposition 2, the sequences $\{x_t, y_t\}_t$ generated by proximal-GDA satisfy*

$$\lim_{t\to\infty} \|x_{t+1} - x_t\| = 0, \quad \lim_{t\to\infty} \|y_{t+1} - y_t\| = 0, \quad \lim_{t\to\infty} \|y_t - y^*(x_t)\| = 0.$$

*Proof.* To prove the first and third items of Corollary 1, summing the inequality of Proposition 2 over $t = 0, 1, ..., T - 1$, we obtain that for all $T \geq 1$,

$$\sum_{t=0}^{T-1} \left[2\|x_{t+1} - x_t\|^2 + \frac{1}{4\kappa^2}(\|y_{t+1} - y^*(x_{t+1})\|^2 + \|y_t - y^*(x_t)\|^2)\right]$$
$$\leq H(z_0) - H(z_T)$$
$$\leq H(z_0) - [\Phi(x_T) + g(x_T)]$$
$$\leq H(z_0) - \inf_{x\in\mathbb{R}^m}\left(\Phi(x) + g(x)\right) < +\infty.$$

Letting $T \to \infty$, we conclude that

$$\sum_{t=0}^{\infty} \left[2\|x_{t+1} - x_t\|^2 + \frac{1}{4\kappa^2}(\|y_{t+1} - y^*(x_{t+1})\|^2 + \|y_t - y^*(x_t)\|^2)\right] < +\infty.$$

Therefore, we must have $\lim_{t\to\infty} \|x_{t+1} - x_t\| = \lim_{t\to\infty} \|y_t - y^*(x_t)\| = 0$.

To prove the second item, note that

$$\|y_{t+1} - y_t\| \leq \|y_{t+1} - y^*(x_t)\| + \|y_t - y^*(x_t)\| \overset{eq.\ (20)}{\leq} (\sqrt{1 - \kappa^{-1}} + 1)\|y_t - y^*(x_t)\| \overset{t}{\to} 0.$$

$\qquad\square$

## D    PROOF OF THEOREM 1

**Theorem 1** (Global convergence). *Let Assumption 1 hold and choose the learning rates $\eta_x \leq \frac{1}{\kappa^3(L+3)^2}$, $\eta_y \leq \frac{1}{L}$. Then, proximal-GDA satisfies the following properties.*

1. *The function value sequence $\{(\Phi + g)(x_t)\}_t$ converges to a finite limit $H^* > -\infty$;*
2. *The sequences $\{x_t\}_t, \{y_t\}_t$ are bounded and have compact sets of limit points. Moreover, $(\Phi + g)(x^*) \equiv H^*$ for any limit point $x^*$ of $\{x_t\}_t$;*
3. *Every limit point of $\{x_t\}_t$ is a critical point of $(\Phi + g)(x)$.*

*Proof.* We first prove some useful results on the Lyapunov function $H(z)$. By Assumption 1 we know that $\Phi + g$ is bounded below and have compact sub-level sets, and we first show that $H(z)$ also satisfies these conditions. First, note that $H(z) = \Phi(x) + g(x) + \left(1 - \frac{1}{4\kappa^2}\right)\|y - y^*(x)\|^2 \geq \Phi(x) + g(x)$. Taking infimum over $x, y$ on both sides we obtain that $\inf_{x,y} H(z) \geq \inf_x \Phi(x) + g(x) > -\infty$. This shows that $H(z)$ is bounded below. Second, consider the sub-level set $\mathcal{Z}_\alpha := \{z = (x, y) : H(z) \leq \alpha\}$ for any $\alpha \in \mathbb{R}$. This set is equivalent to $\{(x, y) : \Phi(x) + g(x) + \left(1 - \frac{1}{4\kappa^2}\right)\|y - y^*(x)\|^2 \leq \alpha\}$. For any point $(x, y) \in \mathcal{Z}_\alpha$, the $x$ part is included in the compact set $\{x : \Phi(x) + g(x) \leq \alpha\}$. Therefore, the $x$ in this set must be compact. Also, the $y$ in this set should also be compact as it is inside the co-coercive function $\|y - y^*(x)\|^2$. Hence, we have shown that $H(z)$ is bounded below and have compact sub-level set.

We first show that $\{(\Phi + g)(x_t)\}_t$ has a finite limit. We have shown in Proposition 2 that $\{H(z_t)\}_t$ is monotonically decreasing. Since $H(z)$ is bounded below, we conclude that $\{H(z_t)\}_t$ has a finite limit $H^* > -\infty$, i.e., $\lim_{t\to\infty}(\Phi + g)(x_t) + \left(1 - \frac{1}{4\kappa^2}\right)\|y_t - y^*(x_t)\|^2 = H^*$. Moreover, since $\|y_t - y^*(x_t)\| \xrightarrow{t} 0$, we further conclude that $\lim_{t\to\infty}(\Phi + g)(x_t) = H^*$.

Next, we prove the second item. Since $\{H(z_t)\}_t$ is monotonically decreasing and $H(z)$ has compact sub-level set, we conclude that $\{x_t\}_t, \{y_t\}_t$ are bounded and hence have compact sets of limit points. Next, we derive a bound on the subdifferential. By the optimality condition of the proximal gradient update of $x_t$ and the summation rule of subdifferential in Corollary 1.12.2 of Kruger (2003), we have

$$\mathbf{0} \in \partial g(x_{t+1}) + \frac{1}{\eta_x}\big(x_{t+1} - x_t + \eta_x \nabla_1 f(x_t, y_t)\big).$$

Then, we obtain that

$$\frac{1}{\eta_x}\big(x_t - x_{t+1}\big) - \nabla_1 f(x_t, y_t) + \nabla\Phi(x_{t+1}) \in \partial(\Phi + g)(x_{t+1}), \tag{24}$$

which further implies that

$$\begin{aligned}
\text{dist}_{\partial(\Phi+g)(x_{t+1})}(\mathbf{0}) &\leq \frac{1}{\eta_x}\|x_{t+1} - x_t\| + \|\nabla_1 f(x_t, y_t) - \nabla\Phi(x_{t+1})\| \\
&= \frac{1}{\eta_x}\|x_{t+1} - x_t\| + \|\nabla_1 f(x_t, y_t) - \nabla_1 f(x_{t+1}, y^*(x_{t+1}))\| \\
&\leq \frac{1}{\eta_x}\|x_{t+1} - x_t\| + L(\|x_{t+1} - x_t\| + \|y^*(x_{t+1}) - y_t\|) \\
&\leq \left(\frac{1}{\eta_x} + L\right)\|x_{t+1} - x_t\| + L\big(\|y^*(x_{t+1}) - y^*(x_t)\| + \|y^*(x_t) - y_t\|\big) \\
&\leq \left(\frac{1}{\eta_x} + L(1 + \kappa)\right)\|x_{t+1} - x_t\| + L\|y^*(x_t) - y_t\|.
\end{aligned}$$

Since we have shown that $\|x_{t+1} - x_t\| \xrightarrow{t} 0, \|y^*(x_t) - y_t\| \xrightarrow{t} 0$, we conclude from the above inequality that $\text{dist}_{\partial(\Phi+g)(x_t)}(\mathbf{0}) \xrightarrow{t} 0$. Therefore, we have shown that

$$\frac{1}{\eta_x}\big(x_{t-1} - x_t\big) - \nabla_1 f(x_{t-1}, y_{t-1}) + \nabla\Phi(x_t) \in \partial(\Phi + g)(x_t),$$

$$\text{and } \frac{1}{\eta_x}\big(x_{t-1} - x_t\big) - \nabla_1 f(x_{t-1}, y_{t-1}) + \nabla\Phi(x_t) \xrightarrow{t} \mathbf{0}. \tag{25}$$

Now consider any limit point $x^*$ of $x_t$ so that $x_{t(j)} \xrightarrow{j} x^*$ along a subsequence. By the proximal update of $x_{t(j)}$, we have

$$g(x_{t(j)}) + \frac{1}{2\eta_x}\|x_{t(j)} - x_{t(j)-1}\|^2 + \langle x_{t(j)} - x_{t(j)-1}, \nabla_1 f(x_{t(j)-1}, y_{t(j)-1})\rangle$$

$$\leq g(x^*) + \frac{1}{2\eta_x}\|x^* - x_{t(j)-1}\|^2 + \langle x^* - x_{t(j)-1}, \nabla_1 f(x_{t(j)-1}, y_{t(j)-1})\rangle.$$

Taking limsup on both sides of the above inequality and noting that $\{x_t\}_t, \{y_t\}_t$ are bounded, $\nabla f$ is Lipschitz, $\|x_{t+1} - x_t\| \xrightarrow{t} 0$ and $x_{t(j)} \to x^*$, we conclude that $\limsup_j g(x_{t(j)}) \leq g(x^*)$. Since $g$ is lower-semicontinuous, we know that $\liminf_j g(x_{t(j)}) \geq g(x^*)$. Combining these two inequalities yields that $\lim_j g(x_{t(j)}) = g(x^*)$. By continuity of $\Phi$, we further conclude that $\lim_j (\Phi + g)(x_{t(j)}) = (\Phi + g)(x^*)$. Since we have shown that the entire sequence $\{(\Phi + g)(x_t)\}_t$ converges to a certain finite limit $H^*$, we conclude that $(\Phi + g)(x^*) \equiv H^*$ for all the limit points $x^*$ of $\{x_t\}_t$.

Next, we prove the third item. To this end, we have shown that for every subsequence $x_{t(j)} \xrightarrow{j} x^*$, we have that $(\Phi + g)(x_{t(j)}) \xrightarrow{j} (\Phi + g)(x^*)$ and there exists $u_t \in \partial(\Phi + g)(x_t)$ such that $u_t \xrightarrow{t} \mathbf{0}$ (by eq. (25)). Recall the definition of limiting sub-differential, we conclude that every limit point $x^*$ of $\{x_t\}_t$ is a critical point of $(\Phi + g)(x)$, i.e., $\mathbf{0} \in \partial(\Phi + g)(x^*)$.

$\square$

## E  PROOF OF THEOREM 2

**Theorem 2** (Variable convergence). *Let Assumption 1 & 2 hold and assume that $H$ has the KŁ geometry. Choose the learning rates $\eta_x \leq \frac{1}{\kappa^3(L+3)^2}$ and $\eta_y \leq \frac{1}{L}$. Then, the sequence $\{(x_t, y_t)\}_t$ generated by proximal-GDA converges to a certain critical point $(x^*, y^*(x^*))$ of $(\Phi + g)(x)$, i.e.,*

$$x_t \xrightarrow{t} x^*, \quad y_t \xrightarrow{t} y^*(x^*).$$

*Proof.* We first derive a bound on $\partial H(z)$. Recall that $H(z) = \Phi(x) + g(x) + \left(1 - \frac{1}{4\kappa^2}\right)\|y - y^*(x)\|^2$, and that $\|y^*(x) - y\|^2$ has non-empty subdifferential $\partial_x(\|y^*(x) - y\|^2)$. We therefore have

$$\partial_x H(z) \supset \partial(\Phi + g)(x) + \left(1 - \frac{1}{4\kappa^2}\right)\partial_x(\|y^*(x) - y\|^2),$$

$$\nabla_y H(z) = -\left(2 - \frac{1}{2\kappa^2}\right)(y^*(x) - y),$$

where the first inclusion follows from the scalar multiplication rule and sum rule of sub-differential, see Proposition 1.11 & 1.12 of Kruger (2003). Next, we derive upper bounds on these sub-differentials. Based on Definition 1, we can take any $u \in \widehat{\partial}_x(\|y^*(x) - y\|^2)$ and obtain that

$$
\begin{aligned}
0 &\leq \liminf_{z \neq x, z \to x} \frac{\|y^*(z) - y\|^2 - \|y^*(x) - y\|^2 - u^\mathsf{T}(z - x)}{\|z - x\|} \\
&\leq \liminf_{z \neq x, z \to x} \frac{[y^*(z) - y^*(x)]^\mathsf{T}[y^*(z) + y^*(x) - 2y] - u^\mathsf{T}(z - x)}{\|z - x\|} \\
&\leq \liminf_{z \neq x, z \to x} \frac{\|y^*(z) - y^*(x)\|\|y^*(z) + y^*(x) - 2y\| - u^\mathsf{T}(z - x)}{\|z - x\|} \\
&\overset{(i)}{\leq} \liminf_{z \neq x, z \to x} \left[\kappa\|y^*(z) + y^*(x) - 2y\| - \frac{u^\mathsf{T}(z - x)}{\|z - x\|}\right] \\
&\overset{(ii)}{=} 2\kappa\|y^*(x) - y\| - \limsup_{z \neq x, z \to x} \frac{u^\mathsf{T}(z - x)}{\|z - x\|} \\
&\overset{(iii)}{=} 2\kappa\|y^*(x) - y\| - \|u\|
\end{aligned}
\tag{26}
$$

where (i) and (ii) use the fact that $y^*$ is $\kappa$-Lipschitz based on Proposition 1, and the limsup in (iii) is achieved by letting $z = x + \sigma u$ with $\sigma \to 0^+$ in (ii). Hence, we conclude that $\|u\| \leq 2\kappa\|y^*(x) - y\|$. Since $\partial_x(\|y^*(x) - y\|^2)$ is the graphical closure of $\widehat{\partial}_x(\|y^*(x) - y\|^2)$, we have that

$$\text{dist}_{\partial_x(\|y^*(x)-y\|^2)}(\mathbf{0}) \leq 2\kappa\|y^*(x) - y\|.$$

Then, utilizing the characterization of $\partial(\Phi + g)(x)$ in eq. (24), we obtain that

$$
\begin{aligned}
&\text{dist}_{\partial H(z_{t+1})}(\mathbf{0}) \\
&\leq \text{dist}_{\partial_x H(z_{t+1})}(\mathbf{0}) + \|\nabla_y H(z_{t+1})\| \\
&\leq \text{dist}_{\partial(\Phi+g)(x_{t+1})}(\mathbf{0}) + \left(1 - \frac{1}{4\kappa^2}\right)\text{dist}_{\partial_x(\|y^*(x_{t+1})-y_{t+1}\|^2)}(\mathbf{0}) + \left(2 - \frac{1}{2\kappa^2}\right)\|y^*(x_{t+1}) - y_{t+1}\| \\
&\leq \frac{1}{\eta_x}\|x_{t+1} - x_t\| + \|\nabla_1 f(x_t, y_t) - \nabla\Phi(x_{t+1})\| + \left(2 - \frac{1}{2\kappa^2}\right)(1 + \kappa)\|y^*(x_{t+1}) - y_{t+1}\| \\
&\overset{(i)}{\leq} \left(\frac{1}{\eta_x} + L\right)\|x_{t+1} - x_t\| + L\|y^*(x_{t+1}) - y_t\| + 2(1 + \kappa)\|y^*(x_{t+1}) - y_{t+1}\| \\
&\overset{(ii)}{\leq} \left(\frac{1}{\eta_x} + L(1 + \kappa)\right)\|x_{t+1} - x_t\| + L\|y^*(x_t) - y_t\| \\
&\qquad + 2(1 + \kappa)\left[\sqrt{1 - \kappa^{-2}}\|y^*(x_t) - y_t\| + \kappa\sqrt{(1 + \kappa)}\|x_{t+1} - x_t\|\right] \\
&\overset{(iii)}{\leq} \left(\frac{1}{\eta_x} + (L + 4\kappa^2)(1 + \kappa)\right)\|x_{t+1} - x_t\| + (L + 4\kappa)\|y^*(x_t) - y_t\|. \qquad (27)
\end{aligned}
$$

where (i) uses Proposition 1 that $\nabla\Phi(x_{t+1}) = \nabla_1 f(x_{t+1}, y^*(x_{t+1}))$ and that $y^*$ is $\kappa$-Lipschitz, (ii) uses eq. (21) and the inequality that $\sqrt{a + b} \leq \sqrt{a} + \sqrt{b}$ $(a, b \geq 0)$ and (iii) uses $\kappa \geq 1$.

Next, we prove the convergence of the sequence under the assumption that $H(z)$ is a KŁ function. Recall that we have shown in the proof of Theorem 1 that: 1) $\{H(z_t)\}_t$ decreases monotonically to the finite limit $H^*$; 2) for any limit point $x^*, y^*$ of $\{x_t\}_t, \{y_t\}_t$, $H(x^*, y^*)$ has the constant value $H^*$. Hence, the KŁ inequality (see Definition 2) holds after sufficiently large number of iterations, i.e., there exists $t_0 \in \mathbb{N}^+$ such that for all $t \geq t_0$,

$$
\varphi'(H(z_t) - H^*)\text{dist}_{\partial H(z_t)}(\mathbf{0}) \geq 1.
$$

Rearranging the above inequality and utilizing eq. (27), we obtain that for all $t \geq t_0$,

$$
\begin{aligned}
&\varphi'(H(z_t) - H^*) \\
&\geq \frac{1}{\text{dist}_{\partial H(z_t)}(\mathbf{0})} \\
&\geq \left[\left(\frac{1}{\eta_x} + (L + 4\kappa^2)(1 + \kappa)\right)\|x_t - x_{t-1}\| + (L + 4\kappa)\|y^*(x_{t-1}) - y_{t-1}\|\right]^{-1} \qquad (28)
\end{aligned}
$$

By concavity of the function $\varphi$ (see Definition 2), we know that

$$
\begin{aligned}
&\varphi(H(z_t) - H^*) - \varphi(H(z_{t+1}) - H^*) \\
&\geq \varphi'(H(z_t) - H^*)(H(z_t) - H(z_{t+1})) \\
&\overset{(i)}{\geq} \frac{\|x_{t+1} - x_t\|^2 + \frac{1}{4\kappa^2}\|y_t - y^*(x_t)\|^2}{\left(\frac{1}{\eta_x} + (L + 4\kappa^2)(1 + \kappa)\right)\|x_t - x_{t-1}\| + (L + 4\kappa)\|y^*(x_{t-1}) - y_{t-1}\|} \qquad (29) \\
&\overset{(ii)}{\geq} \frac{\frac{1}{2}\left[\|x_{t+1} - x_t\| + \frac{1}{2\kappa}\|y_t - y^*(x_t)\|\right]^2}{\left(\frac{1}{\eta_x} + (L + 4\kappa^2)(1 + \kappa)\right)\|x_t - x_{t-1}\| + (L + 4\kappa)\|y^*(x_{t-1}) - y_{t-1}\|},
\end{aligned}
$$

where (i) uses Proposition 2 and eq. (28), (ii) uses the inequality that $a^2 + b^2 \geq \frac{1}{2}(a + b)^2$.

Rearranging the above inequality that

$$
\begin{aligned}
&\left[\|x_{t+1} - x_t\| + \frac{1}{2\kappa}\|y_t - y^*(x_t)\|\right]^2 \\
&\leq 2[\varphi(H(z_t) - H^*) - \varphi(H(z_{t+1}) - H^*)] \\
&\qquad \left[\left(\frac{1}{\eta_x} + (L + 4\kappa^2)(1 + \kappa)\right)\|x_t - x_{t-1}\| + (L + 4\kappa)\|y^*(x_{t-1}) - y_{t-1}\|\right] \\
&\leq \left[C[\varphi(H(z_t) - H^*) - \varphi(H(z_{t+1}) - H^*)]\right.
\end{aligned}
$$

$$+\frac{1}{C}\Big(\frac{1}{\eta_x}+(L+4\kappa^2)(1+\kappa)\Big)\|x_t-x_{t-1}\|+\frac{1}{C}(L+4\kappa)\|y^*(x_{t-1})-y_{t-1}\|\Big]^2$$

where the final step uses the inequality that $2ab \le (Ca+\frac{b}{C})^2$ for any $a,b \ge 0$ and $C > 0$ (the value of $C$ will be assigned later). Taking square root of both sides of the above inequality and telescoping over $t = t_0, \ldots, T-1$, we obtain that

$$\sum_{t=t_0}^{T-1}\|x_{t+1}-x_t\|+\frac{1}{2\kappa}\sum_{t=t_0}^{T-1}\|y_t-y^*(x_t)\|$$

$$\le C\varphi[H(z_{t_0})-H^*]-C\varphi[H(z_T)-H^*]+\frac{1}{C}\Big(\frac{1}{\eta_x}+(L+4\kappa^2)(1+\kappa)\Big)\sum_{t=t_0}^{T-1}\|x_t-x_{t-1}\|$$

$$+\frac{1}{C}(L+4\kappa)\sum_{t=t_0}^{T-1}\|y^*(x_{t-1})-y_{t-1}\|$$

$$\le \frac{Cc}{\theta}[H(z_{t_0})-H^*]^\theta+\frac{1}{C}\Big(\frac{1}{\eta_x}+(L+4\kappa^2)(1+\kappa)\Big)\sum_{t=t_0-1}^{T-2}\|x_{t+1}-x_t\|$$

$$+\frac{1}{C}(L+4\kappa)\sum_{t=t_0-1}^{T-2}\|y^*(x_t)-y_t\|$$

where the final steps uses $\varphi(s) = \frac{c}{\theta}s^\theta$ and the fact that $H(z_T) - H^* \ge 0$. Since the value of $C > 0$ is arbitrary, we can select large enough $C$ such that $\frac{1}{C}\Big(\frac{1}{\eta_x}+(L+4\kappa)^2(1+\kappa)\Big) < \frac{1}{2}$ and $\frac{1}{C}(L+4\kappa) < \frac{1}{2\kappa}$. Hence, the inequality above further implies that

$$\frac{1}{2}\sum_{t=t_0}^{T-1}\|x_{t+1}-x_t\| \le \frac{Cc}{\theta}[H(z_{t_0})-H^*]^\theta+\frac{1}{2}\|x_{t_0}-x_{t_0-1}\|+\frac{1}{2\kappa}\|y^*(x_{t_0-1})-y_{t_0-1}\| < +\infty.$$

Letting $T \to \infty$, we conclude that

$$\sum_{t=1}^{\infty}\|x_{t+1}-x_t\| < +\infty.$$

Moreover, this implies that $\{x_t\}_t$ is a Cauchy sequence and therefore converges to a certain limit, i.e., $x_t \xrightarrow{t} x^*$. We have shown in Theorem 1 that any such limit point must be a critical point of $\Phi + g$. Hence, we conclude that $\{x_t\}_t$ converges to a certain critical point $x^*$ of $(\Phi + g)(x)$. Also, note that $\|y^*(x_t)-y_t\| \xrightarrow{t} 0$, $x_t \xrightarrow{t} x^*$ and $y^*$ is a Lipschitz mapping, so we conclude that $\{y_t\}_t$ converges to $y^*(x^*)$.

$\square$

## F    PROOF OF THEOREM 3

**Theorem 3** (Funtion value convergence rate). *Under the same conditions as those of Theorem 2, the Lyapunov function value sequence $\{H(z_t)\}_t$ converges to the limit $H^*$ at the following rates.*

*1. If KŁ geometry holds with $\theta = 1$, then $H(z_t) \downarrow H^*$ within **finite number of iterations**;*
*2. If KŁ geometry holds with $\theta \in (\frac{1}{2}, 1)$, then $H(z_t) \downarrow H^*$ **super-linearly** as*

$$H(z_t) - H^* \le (2Mc^2)^{-\frac{1}{2\theta-1}}\exp\Big(-\Big(\frac{1}{2(1-\theta)}\Big)^{t-t_0}\Big), \quad \forall t \ge t_0; \qquad (6)$$

*3. If KŁ geometry holds with $\theta = \frac{1}{2}$, then $H(z_t) \downarrow H^*$ **linearly** as*

$$H(z_t) - H^* \le \Big(1+\frac{1}{2Mc^2}\Big)^{t_0-t}(H(z_{t_0})-H^*), \quad \forall t \ge t_0; \qquad (7)$$

4. *If KŁ geometry holds with $\theta \in (0, \frac{1}{2})$, then $H(z_t) \downarrow H^*$ **sub-linearly** as*

$$H(z_t) - H^* \leq \left[ C(t - t_0) \right]^{-\frac{1}{1-2\theta}}, \quad \forall t \geq t_0. \tag{8}$$

*where $C = \min \left[ \frac{1-2\theta}{8Mc^2}, d_{t_0}^{-(1-2\theta)} \left( 1 - 2^{-(1-2\theta)} \right) \right] > 0$.*

*Proof.* Note that eq. (27) implies that

$$\text{dist}_{\partial H(z_{t+1})}(\mathbf{0})^2 \leq 2 \left( \frac{1}{\eta_x} + (L + 4\kappa^2)(1 + \kappa) \right)^2 \|x_{t+1} - x_t\|^2 + 2(L + 4\kappa)^2 \|y^*(x_t) - y_t\|^2, \tag{30}$$

Recall that we have shown that for all $t \geq t_0$, the KŁ property holds and we have

$$\left[ \varphi'(H(z_t) - H^*) \right]^2 \text{dist}_{\partial H(z_t)}^2(\mathbf{0}) \geq 1.$$

Throughout the rest of the proof, we assume $t \geq t_0$. Substituting eq. (30) into the above bound yields that

$$\begin{aligned}
1 \leq & 2 \left[ \varphi'(H(z_t) - H^*) \right]^2 \left[ \left( \frac{1}{\eta_x} + (L + 4\kappa^2)(1 + \kappa) \right)^2 \|x_t - x_{t-1}\|^2 \right. \\
& \left. + (L + 4\kappa)^2 \|y^*(x_{t-1}) - y_{t-1}\|^2 \right]. \\
\leq & 2M \left[ \varphi'(H(z_t) - H^*) \right]^2 \left[ 2\|x_t - x_{t-1}\|^2 + \frac{1}{4\kappa^2} \|y^*(x_{t-1}) - y_{t-1}\|^2 \right] \tag{31}
\end{aligned}$$

where the second inequality uses the definition of $M$ in eq. (5).

Substituting eq. (4) and $\varphi'(s) = cs^{\theta-1}$ $(c > 0)$ into eq. (31) and rearranging, we further obtain that

$$\left[ c(H(z_t) - H^*)^{\theta-1} \right]^{-2} \leq 2M[H(z_{t-1}) - H(z_t)]$$

Defining $d_t = H(z_t) - H^*$, the above inequality further becomes

$$d_{t-1} - d_t \geq \frac{1}{2Mc^2} d_t^{2(1-\theta)}. \tag{32}$$

Next, we prove the convergence rates case by case.

(Case 1) If $\theta = 1$, then eq. (32) implies that $d_{t-1} - d_t \geq \frac{1}{2Mc^2} > 0$ whenever $d_t > 0$. Hence, $d_t$ achieves 0 (i.e., $H(z_t)$ achieves $H^*$) within finite number of iterations.

(Case 2) If $\theta \in (\frac{1}{2}, 1)$, since $d_t \geq 0$, eq. (32) implies that

$$d_{t-1} \geq \frac{1}{2Mc^2} d_t^{2(1-\theta)}, \tag{33}$$

which is equivalent to that

$$(2Mc^2)^{\frac{1}{2\theta-1}} d_t \leq \left[ (2Mc^2)^{\frac{1}{2\theta-1}} d_{t-1} \right]^{\frac{1}{2(1-\theta)}} \tag{34}$$

Since $d_t \downarrow 0$, $(2Mc^2)^{\frac{1}{2\theta-1}} d_{t_1} \leq e^{-1}$ for sufficiently large $t_1 \in \mathbb{N}^+$ and $t_1 \geq t_0$. Hence, eq. (34) implies that for $t \geq t_1$

$$\begin{aligned}
(2Mc^2)^{\frac{1}{2\theta-1}} d_t \leq & \left[ (2Mc^2)^{\frac{1}{2\theta-1}} d_{t_1} \right]^{\left[ \frac{1}{2(1-\theta)} \right]^{t-t_1}} \\
\leq & \exp \left\{ - \left[ \frac{1}{2(1-\theta)} \right]^{t-t_1} \right\}.
\end{aligned}$$

Note that $\theta \in (\frac{1}{2}, 1)$ implies that $\frac{1}{2(1-\theta)} > 1$, and thus the inequality above implies that $H(z_t) \downarrow H^*$ at the super-linear rate given by eq. (6).

(Case 3) If $\theta = \frac{1}{2}$,

$$d_{t-1} - d_t \geq \frac{1}{2Mc^2} d_t, \tag{35}$$

which implies that $d_t \leq \left(1 + \frac{1}{2Mc^2}\right)^{-1} d_{t-1}$. Therefore, $d_t \downarrow 0$ (i.e., $H(z_t) \downarrow H^*$) at the linear rate given by eq. (7).

(Case 4) If $\theta \in (0, \frac{1}{2})$, consider the following two subcases.

If $d_{t-1} \leq 2d_t$, denote $\psi(s) = \frac{1}{1-2\theta} s^{-(1-2\theta)}$, then

$$\psi(d_t) - \psi(d_{t-1}) = \int_{d_t}^{d_{t-1}} -\psi'(s)ds = \int_{d_t}^{d_{t-1}} s^{-2(1-\theta)}ds \overset{(i)}{\geq} d_{t-1}^{-2(1-\theta)}(d_{t-1} - d_t)$$

$$\overset{(ii)}{\geq} \frac{1}{2Mc^2}\left(\frac{d_t}{d_{t-1}}\right)^{2(1-\theta)} \geq \frac{1}{2^{3-2\theta}Mc^2} \geq \frac{1}{8Mc^2} \tag{36}$$

where (i) uses $d_t \leq d_{t-1}$ and $-2(1 - \theta) < -1$, and (ii) uses eq. (32).

If $d_{t-1} > 2d_t$

$$\psi(d_t) - \psi(d_{t-1}) = \frac{1}{1 - 2\theta}(d_t^{-(1-2\theta)} - d_{t-1}^{-(1-2\theta)}) \geq \frac{1}{1 - 2\theta}(d_t^{-(1-2\theta)} - (2d_t)^{-(1-2\theta)})$$

$$\geq \frac{1 - 2^{-(1-2\theta)}}{1 - 2\theta} d_t^{-(1-2\theta)} \geq \frac{1 - 2^{-(1-2\theta)}}{1 - 2\theta} d_{t_0}^{-(1-2\theta)}. \tag{37}$$

where we use $-(1 - 2\theta) < 0$, $d_{t-1} > 2d_t$ and $d_t \leq d_{t_0}$.

Hence,

$$\psi(d_t) - \psi(d_{t-1}) \geq \min\left[\frac{1}{8Mc^2}, \frac{1 - 2^{-(1-2\theta)}}{1 - 2\theta} d_{t_0}^{-(1-2\theta)}\right] = \frac{C}{1 - 2\theta} > 0, \tag{38}$$

which implies that

$$\psi(d_t) \geq \psi(d_{t_0}) + \frac{C}{1 - 2\theta}(t - t_0) \geq \frac{C}{1 - 2\theta}(t - t_0)$$

By substituing the definition of $\psi$, the inequality above implies that $H(z_t) \downarrow H^*$ in a sub-linear rate given by eq. (8). $\qquad\square$

## G   PROOF OF THEOREM 4

**Theorem 4** (Variable convergence rate). *Under the same conditions as those of Theorem 2, the sequences $\{x_t, y_t\}_t$ converge to their limits $x^*, y^*(x^*)$ respectively at the following rates.*

1. *If KŁ geometry holds with $\theta = 1$, then $(x_t, y_t) \to (x^*, y^*(x^*))$ within **finite number of iterations**;*

2. *If KŁ geometry holds with $\theta \in (\frac{1}{2}, 1)$, then $(x_t, y_t) \to (x^*, y^*(x^*))$ **super-linearly** as*

$$\max\left\{\|x_t - x^*\|, \|y_t - y^*(x^*)\|\right\} \leq \mathcal{O}\left(\exp\left(-\left(\frac{1}{2(1-\theta)}\right)^{t-t_0}\right)\right), \quad \forall t \geq t_0; \tag{9}$$

3. *If KŁ geometry holds with $\theta = \frac{1}{2}$, then $(x_t, y_t) \to (x^*, y^*(x^*))$ **linearly** as*

$$\max\left\{\|x_t - x^*\|, \|y_t - y^*(x^*)\|\right\} \leq \mathcal{O}\left(\left(\min\left\{2, 1 + \frac{1}{2Mc^2}\right\}\right)^{(t_0-t)/2}\right), \quad \forall t \geq t_0; \tag{10}$$

4. *If KŁ geometry holds with $\theta \in (0, \frac{1}{2})$, then $(x_t, y_t) \to (x^*, y^*(x^*))$ **sub-linearly** as*

$$\max\left\{\|x_t - x^*\|, \|y_t - y^*(x^*)\|\right\} \leq \mathcal{O}\left((t - t_0)^{-\frac{\theta}{1-2\theta}}\right), \quad \forall t \geq t_0. \tag{11}$$

*Proof.* (Case 1) If $\theta = 1$, then based on the first case of Appendix F, $H(z_t) \equiv H^*$ after finite number of iterations. Hence, for large enough $t$, Proposition 2 yields that

$$2\|x_{t+1} - x_t\|^2 + \frac{1}{4\kappa^2}\left(\|y_{t+1} - y^*(x_{t+1})\|^2 + \|y_t - y^*(x_t)\|^2\right) \leq H(z_t) - H(z_{t+1}) = 0, \quad (39)$$

which implies that $x_{t+1} = x_t$ and $y_t = y^*(x_t)$ for large enougth $t$. Hence, $x_t \to x^*$ and $y_t \to y^*(x^*)$ within finite number of iterations.

(Case 2) If $\theta \in (\frac{1}{2}, 1)$, denote $A_t = \|x_{t+1} - x_t\| + \frac{1}{2\kappa}\|y_t - y^*(x_t)\|$. Then, based on the definition of $M$ in eq. (5), we have

$$\left(\frac{1}{\eta_x} + (L + 4\kappa^2)(1 + \kappa)\right)\|x_t - x_{t-1}\| + (L + 4\kappa)\|y^*(x_{t-1}) - y_{t-1}\| \leq \sqrt{2M}A_{t-1}. \quad (40)$$

Hence, eqs. (28) & (40) and $\varphi'(s) = cs^{\theta-1}$ imply that

$$c(H(z_t) - H^*)^{\theta-1} \geq (\sqrt{2M}A_{t-1})^{-1},$$

which along with $\theta - 1 < 0$ implies

$$H(z_t) - H^* \leq (c\sqrt{2M}A_{t-1})^{\frac{1}{1-\theta}}. \quad (41)$$

Then, eqs. (29) & (40) imply that

$$\varphi(H(z_t) - H^*) - \varphi(H(z_{t+1}) - H^*) \geq \frac{\|x_{t+1} - x_t\|^2 + \frac{1}{4\kappa^2}\|y_t - y^*(x_t)\|^2}{2\sqrt{2M}A_{t-1}}.$$

Using the inequality that $a^2 + b^2 \geq \frac{1}{2}(a + b)^2$ and recalling the definition of $A_t$ and $\varphi(s) = \frac{c}{\theta}s^\theta$, the above inequality further implies that

$$\frac{c}{\theta}(H(z_t) - H^*)^\theta - \frac{c}{\theta}(H(z_{t+1}) - H^*)^\theta \geq \frac{A_t^2}{4\sqrt{2M}A_{t-1}}. \quad (42)$$

Substituting eq. (41) into eq. (42) and using $H(z_{t+1}) - H^* \geq 0$ yield that

$$A_t^2 \leq \frac{4}{\theta}(c\sqrt{2M}A_{t-1})^{\frac{1}{1-\theta}},$$

which is equivalent to that

$$C_1 A_t \leq (C_1 A_{t-1})^{\frac{1}{2(1-\theta)}}, \quad (43)$$

where

$$C_1 = (4/\theta)^{\frac{1-\theta}{2\theta-1}}(c\sqrt{2M})^{\frac{1}{2\theta-1}}.$$

Note that eq. (43) holds for $t \geq t_0$. Since $A_t \to 0$, there exists $t_1 \geq t_0$ such that $C_1 A_{t_1} \leq e^{-1}$. Hence, by iterating eq. (43) from $t = t_1 + 1$, we obtain

$$C_1 A_t \leq \exp\left[-\left(\frac{1}{2(1-\theta)}\right)^{t-t_1}\right], \quad \forall t \geq t_1 + 1.$$

Hence, for any $t \geq t_1 + 1$,

$$\sum_{s=t}^{\infty} A_s \leq \frac{1}{C_1}\sum_{s=t}^{\infty}\exp\left[-\left(\frac{1}{2(1-\theta)}\right)^{s-t_1}\right]$$

$$= \frac{1}{C_1}\exp\left[-\left(\frac{1}{2(1-\theta)}\right)^{t-t_1}\right]\sum_{s=t}^{\infty}\exp\left[\left(\frac{1}{2(1-\theta)}\right)^{t-t_1} - \left(\frac{1}{2(1-\theta)}\right)^{s-t_1}\right]$$

$$= \frac{1}{C_1}\exp\left[-\left(\frac{1}{2(1-\theta)}\right)^{t-t_1}\right]\sum_{s=t}^{\infty}\exp\left\{\left(\frac{1}{2(1-\theta)}\right)^{t-t_1}\left[1 - \left(\frac{1}{2(1-\theta)}\right)^{s-t}\right]\right\}$$

$$\overset{(i)}{\leq} \frac{1}{C_1} \exp\left[-\left(\frac{1}{2(1-\theta)}\right)^{t-t_1}\right] \sum_{s=t}^{\infty} \exp\left[1-\left(\frac{1}{2(1-\theta)}\right)^{s-t}\right]$$

$$= \frac{1}{C_1} \exp\left[-\left(\frac{1}{2(1-\theta)}\right)^{t-t_1}\right] \sum_{s=0}^{\infty} \exp\left[1-\left(\frac{1}{2(1-\theta)}\right)^{s}\right]$$

$$\overset{(ii)}{\leq} \mathcal{O}\left\{\exp\left[-\left(\frac{1}{2(1-\theta)}\right)^{t-t_1}\right]\right\}, \tag{44}$$

where (i) uses the inequalities that $\frac{1}{2(1-\theta)} > 1$ and that $s \geq t \geq t_1 + 1$, and (ii) uses the fact that $\sum_{s=0}^{\infty} \exp\left[1-\left(\frac{1}{2(1-\theta)}\right)^{s}\right] < +\infty$ is a positive constant independent from $t$. Therefore, the convergence rate (9) can be directly derived as follows

$$\|x_t - x^*\| = \limsup_{T \to \infty} \|x_t - x_T\| \leq \limsup_{T \to \infty} \sum_{s=t}^{T-1} \|x_{s+1} - x_s\|$$

$$\leq \limsup_{T \to \infty} \sum_{s=t}^{T-1} A_s \leq \mathcal{O}\left\{\exp\left[-\left(\frac{1}{2(1-\theta)}\right)^{t-t_1}\right]\right\}, \tag{45}$$

and

$$\|y_t - y^*(x^*)\| \leq \|y_t - y^*(x_t)\| + \|y^*(x_t) - y^*(x^*)\| \overset{(i)}{\leq} 2\kappa A_t + \kappa\|x_t - x^*\|$$

$$\leq 2\kappa \sum_{s=t}^{\infty} A_s + \kappa\|x_t - x^*\| \overset{(ii)}{\leq} \mathcal{O}\left\{\exp\left[-\left(\frac{1}{2(1-\theta)}\right)^{t-t_1}\right]\right\},$$

where (i) uses the Lipschitz property of $y^*$ in Proposition 1, and (ii) uses eqs. (44) & (45).

(Case 3 & 4) Notice that eq. (42) still holds if $\theta \in \left(0, \frac{1}{2}\right]$. Hence, if $A_t \geq \frac{1}{2}A_{t-1}$, then eq. (42) implies that

$$A_t \leq \frac{8c\sqrt{2M}}{\theta}\left[(H(z_t) - H^*)^\theta - (H(z_{t+1}) - H^*)^\theta\right].$$

Otherwise, $A_t \leq \frac{1}{2}A_{t-1}$. Combining these two inequalities yields that

$$A_t \leq \frac{8c\sqrt{2M}}{\theta}\left[(H(z_t) - H^*)^\theta - (H(z_{t+1}) - H^*)^\theta\right] + \frac{1}{2}A_{t-1}.$$

Notice that the inequality above holds whenever $t \geq t_0$. Hence, telescoping the inequality above yields

$$\sum_{s=t}^{T} A_s \leq \frac{8c\sqrt{2M}}{\theta}\left[(H(z_t) - H^*)^\theta - (H(z_{T+1}) - H^*)^\theta\right] + \frac{1}{2}\sum_{s=t-1}^{T-1} A_s, \quad \forall t \geq t_0, \tag{46}$$

which along with $A_T \geq 0$, $H(z_{T+1}) - H^* \geq 0$ implies that

$$\frac{1}{2}\sum_{s=t}^{T} A_s \leq \frac{8c\sqrt{2M}}{\theta}(H(z_t) - H^*)^\theta + \frac{1}{2}A_{t-1},$$

Letting $t = t_0$ and $T \to \infty$ in the above inequality yields that $\sum_{s=t_0}^{\infty} A_s < +\infty$. Hence, by letting $T \to \infty$ and denoting $S_t = \sum_{s=t}^{\infty} A_s$ in eq. (46), we obtain that

$$S_t \leq \frac{8c\sqrt{2M}}{\theta}(H(z_t) - H^*)^\theta + \frac{1}{2}S_{t-1}, \quad \forall t \geq t_0,$$

which further implies that

$$S_t \leq \frac{1}{2^{t-t_0}}S_{t_0} + \frac{8c\sqrt{2M}}{\theta}\sum_{s=t_0+1}^{t}\frac{1}{2^{t-s}}(H(z_s) - H^*)^\theta \tag{47}$$

(Case 3) If $\theta = 1/2$, eq. (7) holds. Substituting eq. (7) and $\theta = 1/2$ into eq. (47) yields that

$$S_t \le \frac{1}{2^{t-t_0}} S_{t_0} + 8c\sqrt{2M[H(z_{t_0}) - H^*]} \sum_{s=t_0+1}^{t} \frac{1}{2^{t-s}}\left(1 + \frac{1}{2Mc^2}\right)^{(t_0-s)/2}$$

$$\le \frac{1}{2^{t-t_0}} S_{t_0} + \frac{C_2}{2^t} \sum_{s=t_0+1}^{t} \left(\frac{1}{4} + \frac{1}{8Mc^2}\right)^{-s/2} \tag{48}$$

where

$$C_2 = 8c\sqrt{2M[H(z_{t_0}) - H^*]}\left(1 + \frac{1}{2Mc^2}\right)^{t_0/2} \tag{49}$$

is a positive constant independent of $t$.

Notice that when $\frac{1}{4} + \frac{1}{8Mc^2} \ge 1$,

$$\sum_{s=t_0+1}^{t} \left(\frac{1}{4} + \frac{1}{8Mc^2}\right)^{-s/2} \le t - t_0$$

and when $\frac{1}{4} + \frac{1}{8Mc^2} < 1$,

$$\sum_{s=t_0+1}^{t} \left(\frac{1}{4} + \frac{1}{8Mc^2}\right)^{-s/2} = \left(\frac{1}{4} + \frac{1}{8Mc^2}\right)^{-t/2} \frac{1 - \left(\frac{1}{4} + \frac{1}{8Mc^2}\right)^{(t-t_0)/2}}{1 - \left(\frac{1}{4} + \frac{1}{8Mc^2}\right)^{1/2}} \le \mathcal{O}\left[\left(\frac{1}{4} + \frac{1}{8Mc^2}\right)^{-t/2}\right]$$

Since either of the two above inequalities holds, combining them yields that

$$\sum_{s=t_0+1}^{t} \left(\frac{1}{4} + \frac{1}{8Mc^2}\right)^{-s/2} \le \mathcal{O}\left\{ \max\left[t - t_0, \left(\frac{1}{4} + \frac{1}{8Mc^2}\right)^{-t/2}\right]\right\}$$

Substituing the above inequality into eq. (48) yields that

$$S_t \le \frac{1}{2^{t-t_0}} S_{t_0} + \mathcal{O}\left\{ \max\left[2^{-t}(t - t_0), \left(1 + \frac{1}{2Mc^2}\right)^{-t/2}\right]\right\}$$

$$\le \mathcal{O}\left\{\left[\min\left(2, 1 + \frac{1}{2Mc^2}\right)\right]^{-t/2}\right\}.$$

Hence,

$$\|x_t - x^*\| \overset{(i)}{\le} \sum_{s=t}^{\infty} A_s = S_t \le \mathcal{O}\left\{\left[\min\left(2, 1 + \frac{1}{2Mc^2}\right)\right]^{-t/2}\right\},$$

where (i) comes from eq. (45). Then,

$$\|y_t - y^*(x^*)\| \le \|y_t - y^*(x_t)\| + \|y^*(x_t) - y^*(x^*)\| \le 2\kappa A_t + \kappa\|x_t - x^*\|$$

$$\le 2\kappa S_t + \kappa\|x_t - x^*\| \le \mathcal{O}\left\{\left[\min\left(2, 1 + \frac{1}{2Mc^2}\right)\right]^{-t/2}\right\}.$$

The two above inequalities yield the linear convergence rate (10).

(Case 4) If $\theta \in (0, \frac{1}{2})$, then eq. (8) holds. Substituting eq. (8) into eq. (47) yields that for some constant $C_3 > 0$,

$$S_t \le \frac{1}{2^{t-t_0}} S_{t_0} + \frac{8c\sqrt{2M}}{\theta} \sum_{s=t_0+1}^{t} \frac{C_3}{2^{t-s}}(s - t_0)^{-\frac{\theta}{1-2\theta}}$$

$$\le \frac{1}{2^{t-t_0}} S_{t_0} + \frac{8cC_3\sqrt{2M}}{2^{t-t_0}\theta} \sum_{s=1}^{t-t_0} 2^s s^{-\frac{\theta}{1-2\theta}}$$

$$\overset{(i)}{=} \frac{1}{2^{t-t_0}} S_{t_0} + \frac{8cC_3\sqrt{2M}}{2^{t-t_0}\theta} \sum_{s=1}^{t_1} 2^s s^{-\frac{\theta}{1-2\theta}} + \frac{8cC_3\sqrt{2M}}{2^{t-t_0}\theta} \sum_{s=t_1+1}^{t-t_0} 2^s s^{-\frac{\theta}{1-2\theta}}$$

$$\leq \frac{1}{2^{t-t_0}} S_{t_0} + \frac{8cC_3\sqrt{2M}}{2^{t-t_0}\theta} \sum_{s=1}^{t_1} 2^s + \frac{8cC_3\sqrt{2M}}{2^{t-t_0}\theta} \sum_{s=t_1+1}^{t-t_0} 2^s \left(\frac{t-t_0}{2}\right)^{-\frac{\theta}{1-2\theta}}$$

$$\overset{(ii)}{\leq} \frac{1}{2^{t-t_0}} S_{t_0} + \frac{8cC_3\sqrt{2M}}{2^{t-t_0}\theta} 2^{t_1+1} + \frac{8cC_3\sqrt{2M}}{2^{t-t_0}\theta} \left(\frac{t-t_0}{2}\right)^{-\frac{\theta}{1-2\theta}} 2^{t-t_0+1}$$

$$= \mathcal{O}\left[\frac{1}{2^{t-t_0}} + \frac{1}{2^{(t-t_0)/2}} + (t-t_0)^{-\frac{\theta}{1-2\theta}}\right] = \mathcal{O}\left[(t-t_0)^{-\frac{\theta}{1-2\theta}}\right], \tag{50}$$

where (i) denotes $t_1 = \lfloor (t-t_0)/2 \rfloor$, (ii) uses the inequality that $\sum_{s=t_1+1}^{t-t_0} 2^s < \sum_{s=0}^{t-t_0} 2^s < 2^{t-t_0+1}$. Therefore, the sub-linear convergence rate eq. (11) follows from the following inequalities.

$$\|x_t - x^*\| \leq S_t \leq \mathcal{O}\left[(t-t_0)^{-\frac{\theta}{1-2\theta}}\right],$$

and

$$\|y_t - y^*(x^*)\| \leq \|y_t - y^*(x_t)\| + \|y^*(x_t) - y^*(x^*)\| \leq 2\kappa A_t + \kappa\|x_t - x^*\|$$

$$\leq 2\kappa S_t + \kappa\|x_t - x^*\| \leq \mathcal{O}\left[(t-t_0)^{-\frac{\theta}{1-2\theta}}\right].$$

$\square$

