# OpenReview forum: "Proximal Gradient Descent-Ascent: Variable Convergence under KŁ Geometry"
_ICLR.cc/2021/Conference — ICLR 2021 Poster_

### Official Review · AnonReviewer1 · 2020-10-25
**The results are nice and well-explained; the paper could be improved by adding experiments and better explanation of its relevance to ML**

**Rating:** 7
**Confidence:** 3

**Review:**

## Summary
This paper discusses the convergence of the proximal gradient algorithm in the case of KL geometry. It does not propose a new algorithm but rather focuses on understanding the simplest (although not the best) algorithm that can be applied to the saddle point objective.

I enjoyed reading this paper, I find the theory to be quite interesting and well-grounded. The minmax problem has gained considerable attention recently but is not yet fully developed, and this work has a valid contribution. I think its main strength is the simplicity of the discussed algorithm and I hope that it will lead to further extension. The work is purely theoretical, but I still think it could benefit from some numerical experiments, especially if the authors identified a practical setting that can be described by the paper's assumptions.  The other weakness is the lack of clear motivation: the authors consider a very specific setup, with two proximable functions and one non-bilinear joint term, certain assumptions about each of the functions, and I don't feel that the choices were sufficiently motivated. Still, I vote for acceptance because, in my opinion, the topic is underexplored.

## Additional comments

1. The authors wrote in Section 1.1 that they "propose a proximal-GDA algorithm", and I find this choice of words to be confusing: this algorithm is by far not new as it is a well-known special case of the Forward-Backward iteration, which dates back to (1979, "Splitting algorithms for the sum of two nonlinear operators"). I expect the authors to properly cite the related literature on Forward-Backaward and mention the guarantees that follow from the existing bounds on Forward-Backward algorithm (which has been quite extensively studied).

2. The Lyapunov function in Proposition 2 is quite interesting, but the quadratic term is not really explained. I think the paper would benefit from explaining why this term shows up in the proofs.

3. The bounds in Theorem 1 and other results seem to have a terrible dependency on the conditioning. In the strongly monotone case, the stepsize in GDA has to be of order O(1/(kappa*L)), but here it is O(1/(kappa^3*L)). This is not completely unexpected due to the nonconvexity, but I'm wondering if the authors think that the bounds are tight or they can be improved.

4. The bad conditioning in the bounds seems to be coming from the update of GDA, and a better update should lead to faster rates. I think that the extragradient algorithm should have the right conditioning with stepsize O(1/L), and several methods have been proposed recently that achieve O(kappa) bounds: 1) "Reducing noise in GAN training with variance reduced extragradient", 2) "Revisiting stochastic extragradient", 3) "On the convergence of single-call stochastic extra-gradient methods". I'd appreciate if the authors could comment on that.

5. The authors wrote "The Kurdyka-Łojasiewicz (KŁ) geometry <...> has been shown to hold ubiquitously for most practical functions" -- I think this is a big overstatement,  I haven't seen any example that would be relevant to the minmax optimization.

6. The authors assume that y*(x) is differentiable in Section 4. First of all, please make this assumption more explicit as it's very easy to miss, particularly so because the other assumptions are presented in Assumption 1. Secondly, I'm wondering if the assumption that y*(x) is differentiable is really necessary. The authors prove that y* is Lipschitz, so assuming differentiability makes the class of relevant objectives small. Secondly, as far as I can see, differentiability is only used in the proof of theorem 2, and the authors only need f1(x)=||y-y*(x)||^2 to be differentiable and to satisfy ||nabla f1(x)|| <= const * f1(x). It might be possible to relax the assumption a little bit (for example ||x||^2 is differentiable everywhere but ||x|| isn't).

7. Remark 1 is not precisely correct: to relax the assumption, function h would have to be strongly concave with constant mu>L as otherwise adding gamma*h to f with gamma<1 will not make f strongly concave.

### Minor issues
Abstract: "it is lack of understanding" -> "it lacks understanding"
p.2, "the entire variable sequences of proximal-GDA have a unique limit point" -> "sequence ... has"
p.6, "proximal-GDA admits a very important Lyapunov function" -- it's a bit strange to call the Lyapunov function "very important"

---

> ### Author Response · Authors · 2020-11-18
> **Response**
>
> We thank the reviewer for providing valuable feedback that helps us improve the quality of this paper. Below is a point-to-point response to the questions raised by the reviewer.
>
> Q1: "propose a proximal-GDA algorithm" looks confusing.
>
> A: We agree with the reviewer that the basic updates of the algorithm originates from the forward-backward splitting (Lions and Mercier, 1979). We have changed the wording to "we study a proximal-GDA algorithm that leverages the forward-backward splitting updates", and we have cited and discussed this related literature in the contribution subsection and Section 3.
>
> Q2: The quadratic term in the Lyapunov function is not really explained.
>
> A: To briefly explain, the eq.(19) in the supplementary establishes a recursive inequality on the objective function $(\Phi + g)(x_{t+1})$. One can see that the right hand side of this inequality contains a desired negative term $-||x_{t+1} - x_t||^2$ and an undesired positive term $ ||y^*(x_t) - y_t||^2$. Hence, the objective function $(\Phi + g)(x_{t+1})$ cannot serve as a proper Lyapunov function. In the subsequent analysis, we break the positive term into a difference term $(1-1/2\kappa^2)||y^*(x_t) - y_t||^2 - ||y^*(x_{t+1}) - y_{t+1}||^2$ by leveraging the $y$-update on the strongly-concave function. After proper rearranging, this difference term contributes to the quadratic term in the final Lyapunov function. We have included this explanation after Proposition 2 in the revision.
>
> Q3: Stepsize and bounds have bad dependency on the conditioning number.
>
> A: We agree with the reviewer on this. These stepsize choices are used in order to construct the Lyapunov function. Specifically, in eqs.(19, 21), we use several AM-GM inequalities that lead to the undesired bad dependence on the condition number. Currently, we are not able to fully resolve this issue, but we think it may be possible to improve these bounds by optimizing the applications of the AM-GM inequalities (fine-tune the associated coefficients).
>
> Q4: The authors wrote "The Kurdyka-Łojasiewicz (KŁ) geometry has been shown to hold ubiquitously for most practical functions" This is a big overstatement.
>
> A: We agree. We have changed this to "The Kurdyka-Łojasiewicz (KŁ) geometry has been shown to hold for a broad class of functions".
>
> Q5: Differentiability of $y^*(x)$ can be relaxed?
>
> A: We thank the reviewer for pointing out this. We realized that it suffices to assume subdifferentiability of the quadratic term $||y^*(x)-y||^2$, see Assumption 2 in the revision. With this weak assumption, we prove in Section E that the subdifferential satisfies the bound $dist_{\partial_x ||y^*(x)-y||^2}(\mathbf{0}) \le 2\kappa ||y^*(x)-y||$. The proof idea is to leverage the definition of the subdifferential to establish an upper bound for it.
>
> Q6: Remark 1 is not precisely correct.
>
> A: We thank the reviewer for pointing out this. We have corrected it correspondingly.
>
> We also greatly thank the reviewer for pointing out the minor issues. We have corrected them in the revision.

---

> > ### Comment · AnonReviewer1 · 2020-11-24
> > **Thank you for addressing my concerns**
> >
> > I am glad to see that the presentation is somewhat improved. I remain positive on that the paper should be accepted. I could increase my score if the dependence on $\kappa$ was improved, but it is not clear if the amendments proposed by the authors are easy to do.
> > I also read the review by Reviewer 4, which explains that the analysis is trivial and has little to offer on top of existing works of Lin et al. and the literature on Kurdyka-Lojasiewicz property. I think that this issue is not that significant since the problem class is of interest even if the theory builds significantly on top of existing works.

---

### Official Review · AnonReviewer4 · 2020-10-27
**Review on "Proximal Gradient Descent-Ascent: Variable Convergence under KŁ Geometry"**

**Rating:** 5
**Confidence:** 4

**Review:**

In this paper the authors analyze a proximal gradient descent-ascent method for nonconvex minimax problem under specific assumptions (another name would be a forward-backward algorithm). The authors prove subsequence convergence of iterates to critical points and furthermore,  additionally under the Kurdyka-Łojasiewicz condition, prove iterate convergence with some  rates.

I have two main concerns.
1) The first one is that required assumptions are quite restrictive. Basically, whatever analysis needs the authors included into assumptions. There is no much discussion about Assumption 1. I understand that it is just a concatenation of different and popular assumptions in the literature, but together they look stringent. What are the real interesting examples that satisfy all of them?

2) The second concern is that a big part of obtained results is very natural if not known already. The authors cite Lin et al. (2020), where the problem $\min_x \max_y f(x, y)$ was studied. Adding regularizers $g$ and $h$ do not complicate things much. Even though we allow $g$ to be nonconvex, we use a proximal step for it, which preserves monotonicity of the Lyapunov energy.  To explain better what I mean, see Attouch,  et. al. "Convergence of descent methods for semi-algebraic and tame problems": once we establish convergence for a forward method (gradient of $f$), adding a backward step (prox of $g$) is easy.
Finally, adding Kurdyka-Łojasiewicz assumption does not change situation much. Its application is standard, once we have some decrease in the objective or energy.


To conclude, I don't see much value in the proposed analysis, since its strength lies in the required assumptions and not in the novel technique.


Below I collect a more detailed list of issues I found.


1. page 1: What is the definition of "function geometry"? Why is it strong convexity/concavity?

2. page 2, "converges to a certain stationary point at a sublinear rate, i.e., $|G(x_t)|\to 0$": I didn't understand such explanation of the sublinear rate.

3. page 2. What is "variable convergence"? Probably the authors mean "iterates convergence".

4. page 2, "The Kurdyka-Łojasiewicz (KŁ) geometry provides a universal characterization...": I am afraid, this is not true, it does not provide a universal characterization.

5. page 2, "a very novel Lyapunov function": Adding more adjectives would only help of course.

6. page 5, "we propose the following proximal-GDA algorithm...": It is a well known method. Another name for it is "forward-backward method".

7. page 5: If $g$ is not convex, how is $x_{t+1}$ uniquely defined?

8. Proposition 1, proof. $\Phi + g$ is lower bounded, but $g$ can be $+\infty$ on the set $A_n$, thus we cannot conclude that $\Phi$ is lower bounded on $A_n$

Proposition 2, proof.
1. page 12: $y^*(x_t)$ is the unique minimizer of the strongly concave function $f(x_t, y) - h(y)$, not of $f(x_t, y)$.
2. page 14: In Eq. 24 it is worth to add a reference for the sum rule of a subdifferential.
3. page 14, Eq. 25: the notation in the end is incorrect (as well as in page 4). Subdifferential is a set.

Theorem 2, proof.
1. page 15. Why is $y^*(x)$ differentiable? We proved that it is only Lipschitz. Note that the sum rule applied to the subdifferential of $H(z)$ won't be valid.

---

> ### Author Response · Authors · 2020-11-18
> **Response**
>
> We thank the reviewer for providing valuable feedback that helps us improve the quality of this paper. Below is a point-to-point response.
>
> Main concerns:
>
> Q1: Assumption 1 is restrictive.
>
> A: Our Assumption 1 is standard and is not a concatenation of existing ones. To elaborate, items (1) and (4) specify the objective function class and regularizer class, which have been widely studied in the existing literature (Lin et al. 2020, Xu et al. 2020). For items (2) and (3), they guarantee that the minimax problem has at least one solution, otherwise it is not well-defined to study variable sequence convergence. These assumptions may not show up in some previous studies, because those studies do not analyze the convergence of variable sequence.
>
> These assumptions are satisfied by many nonconvex-strongly-concave minimax problems, e.g., training of empirical Wasserstein robustness model (Lin et al. 2020), generative adversarial imitation learning of linear quadratic regulators (Nouiehed et al. 2019). In particular, item (4) allows us to add a large class of regularizers to these problems.
>
> Q2: A big part of the obtained results are very natural if not known already.
>
> A: To us, obtaining these results are non-trivial, nor could they be expected in a straightforward manner. To clarify, our contribution is to establish the formal convergence of the variable sequences generated by proximal-GDA, not introducing the regularizers to the problem. Even without the regularizers, this variable convergence result is new for GDA in nonconvex minimax optimization, and is stronger than the gradient norm convergence result established in the existing literature (Lin et al. 2020). We are not aware of any such type of guarantee in the previous studies of GDA in the nonconvex setting.
>
> Regarding our analysis of proximal-GDA under KL geometry, it cannot directly follow from the existing analysis of gradient descent under KL. In the existing studies, KL geometry is exploited to study the variable sequence convergence of gradient-based algorithms for solving minimization problems. In comparison, our work is the first that exploits KL geometry to study the variable convergence of GDA-type algorithms, which are different from and considerably more challenging to analyze than gradient-type algorithms. Specifically, we need to identify the Lyapunov function $H(z)$ of proximal-GDA, which already provides a new perspective into understanding GDA-type algorithms. This special Lyapunov function is different from that associated with the gradient descent (Attouch et al, 2010), which is the objective function of the minimization problem.
>
> Minor issues
>
> Q: Issues 1-8:
>
> A: 1 Function geometry corresponds to the geometry condition that is satisfied by the objective function.
>
> 2 Sorry for the confusion. We have changed it to "i.e., $\|G(x_t)\|\le t^{-\alpha}$ for some $\alpha>0$".
>
> 3 Variable convergence corresponds to the convergence of the variable sequences $x_t,y_t$.
>
> 4 We agree with the reviewer that this is an overclaim. We have changed it from "universal characterization" to "broad characterization".
>
> 5 We deleted the word "very".
>
> 6 Thanks for pointing out this. We have cited the original forward-backward splitting paper (Lions and Mercier 1979).
>
> 7 For nonconvex $g$, its proximal mapping in general can be a set-valued mapping and therefore $x_{t+1}$ need not be uniquely defined. To clarify, $x_{t+1}$ is arbitrarily chosen from the solution set of the proximal subproblem, i.e., $x_{t+1} \in prox_{\eta_x g}(x_t - \eta_x \nabla_1 f(x_t, y_t))$. We have corrected this typo in the update rule of Algorithm 1 in the revision. We note that this does not affect our proof.
>
> 8 We thank the reviewer for pointing out this. We realize that it is unnecessary to use the property of $g$ in that proof. We have fixed this issue and updated the proof in the revision.
>
> To briefly elaborate, note that by definition $\Phi(x)=\max_y f(x,y)-h(y)$. Here, $f(x,y)$ is finite everywhere since it is a smooth function, and $h(y_0)<+\infty$ for at least one $y_0$ since it is a proper convex function. Therefore, for any $x$, we have that $f(x,y_0)-h(y_0)>-\infty$, and then we conclude that $\Phi(x)=\max_y f(x,y)-h(y)>f(x,y_0)-h(y_0) > -\infty$. This further implies that $h(y^*(x)) = f(x, y^*(x)) - \Phi(x) < +\infty$. The rest of the proof is the same as before.
>
> Q: Proposition 2 and Theorem 2 proof.
>
> A: 1 Thanks for pointing out. We have corrected that.
>
> 2 Reference added.
>
> 3 We have rewritten these in the revision. To clarify, in the definition of limiting subdifferential, one needs to identify a sequence of vectors $u_k$ such that 1) $u_k \in \hat{\partial} h(x_k)$ and 2) $u_k \to u$.
>
> 4 Our original proof assumes $y^*(x)$ is differentiable. As suggested by Reviewer 1, we have relaxed the differentiability assumption of $y^*(x)$ to sub-differentiability (see Assumption 2 in revision). The proof in Section E of supplementary is also updated correspondingly.

---

### Official Review · AnonReviewer3 · 2020-10-27
**This paper studied the convergence properties of the proximal-GDA algorithm for solving nonconvex-strongly-concave optimization problems. It develops a novel and comprehensive theoretical understanding of the variable convergence and rates of the algorithm under the KL geometry by identifying an important and intrinsic Lyapunov function.**

**Rating:** 8
**Confidence:** 5

**Review:**

This paper studied the convergence properties of the proximal-GDA algorithm for solving nonconvex-strongly-concave optimization problems. It develops a novel and comprehensive theoretical understanding of the variable convergence and rates of the algorithm under the KL geometry by identifying an important and intrinsic Lyapunov function.

Specifically, the paper considers a regularized nonconvex-strongly-concave optimization problem, with one convex regularizer and another possibly nonconvex and lower-semicontinuous regularizer. This problem formulation generalizes many existing differentiable minimax problems. Then, under standard conditions in Assumption 1, the authors identified a novel and important Lyapunov function H(z) and showed that this function monotonically decreases throughout the optimization process, although the minmax objective function value may oscillate. Based on this new characterization of Lyapunov function, the authors proved that every limit point of the algorithm is a critical point of the minmax problem. Moreover, under the general KL geometry of the Lyapunov function, they formally proved that proximal-GDA converges to a single critical point. This is the first variable convergence result in nonconvex minmax optimization. Besides, they also characterized the dependence of the variable and function value convergence rates on the parameterization of the KL geometry.

Overall I believe this is a novel and important work in minimax optimization. In particular, the Lyapunov function is a very powerful tool for studying the convergence properties of the two variable sequences of proximal-GDA. It conveniently simplifies the analysis of minmax optimization into the analysis of min optimization via Proposition 2. Also, the monotonic decreasing property of the Lyapunov function further enables analyzing this algorithm under the general KL geometry, and obtain a fundamental variable convergence (rates) result(s). These technical tools develop a new framework for analyzing minmax optimization algorithms and can potentially used to study other minmax algorithms.

Below are some of my minor comments:
In Def.2, the {h_\Omega < f(z) < h_\Omega + \lambda}, should be h(z).The proof of Theorem 2 assumes that H is a KL function. As H is essentially the objective function regularized by some quadratic terms, is it sufficient to assume \Phi+g is KL?If possible, I suggest do a simple experiment to verify the monotonic decreasing property of the Lyapunov function.

---

> ### Author Response · Authors · 2020-11-18
> **Response**
>
> We thank the reviewer for providing valuable feedback that helps us improve the quality of this paper. Below is a point-to-point response to the questions raised by the reviewer.
>
> Q1: In Def.2, the ${h_{\Omega} < f(z) < h_{\Omega} + \lambda}$, should be $h(z)$.
>
> A: Yes, and we have fixed this typo in the revision. Thanks a lot.
>
> Q2: The proof of Theorem 2 assumes that H is a KL function. As H is essentially the objective function regularized by some quadratic terms, is it sufficient to assume $\Phi+g$ is KL?
>
> A: No. Note that the function $H(z)$ involves the special quadratic term $||y - y^*(x)||^2$, which depends on $y$ and the mapping $y^*$. Therefore, we need to assume both $(\Phi+g)(x)$ and $||y - y^*(x)||^2$ to be KL.

---

### Official Review · AnonReviewer2 · 2020-10-28
**Proximal Gradient Descent-Ascent: Variable Convergence Under KL Geometry**

**Rating:** 8
**Confidence:** 4

**Review:**

In this paper, the authors analyze the convergence of a proximal gradient descent ascent (GDA) method when applied to non-convex strongly concave functions. To establish convergence results, the authors show that proximal-GDA admits a novel Lyapunov function that monotonically decreases at every iteration. Along with KL-parametrized local geometry, the Lyapunov function was used to establish the convergence of decision variables to a critical point. Moreover, the rate of convergence of the algorithm was computed for various ranges of KL-parameter.

Pros:
The paper studies an interesting and relevant problem in a vibrant field of research.


The convergence analysis of proximal-GDA are detailed and well presented (covering function value convergence, variable convergence, function value convergence rates, and variable value convergence rates). To show the results, the authors provided a novel Lyapunov function and analyzed the convergence using KL local geometry.

The paper is very clear, concise and neatly written (almost free of typos).

The related material are referenced and well-discussed in the paper. The authors clearly positioned their work in the related field and discussed their contributions in comparison to other similar works.


Cons:
The paper lacks any experimental results. Demonstrating the different convergence rates on critical points with various KL-parameter can be good experiment.

Minor Comments:
1.	Definition 2: Should it be $h(z)$ instead of $f(z)$?
2.	Equation (19) in the Appendix, and P Third Inequality should be =.
3.	Page 14: upper bound on distance of subgradient set to 0, second inequality should be =.
4.	Appendix F: Theorem 3, Function missing a ``'c'
5.	Equation (36): $d_{t-1}$ instead of $t_{t-1}$.
6.	Last expression of Page 19: I think 1/2 is missing in the left hand-side.

---------------------------------------------------------------------------------------------------
Satisfied with the response, will keep my score the same.

---

> ### Author Response · Authors · 2020-11-18
> **Response**
>
> We thank the reviewer for pointing out the typos that help improve the quality of this paper, and we have fixed them in the revision. Below are some further clarifications.
>
> Q1: Definition 2: Should it be $h(z)$ instead of $f(z)$?
>
> A: Yes, and thanks for pointing out this typo.
>
> Q2: Equation (19) in the Appendix, and the Third Inequality should be $=$. Page 15: upper bound on distance of subgradient set to 0, second inequality should be $=$.
>
> A: Corrected, and thanks for pointing these out.
>
> Q3: Appendix F: Theorem 3, Function missing a $c$
>
> A: We have explicitly expressed the constant coefficient in the final convergence rate bound in case 4.
>
> Q4: Equation (36): $d_{t-1}$ instead of $t_{t-1}$. Last expression of Page 19: I think $\frac{1}{2}$ is missing in the left hand-side.
>
> A: We have corrected these typos. Thanks a lot.

---

### Decision · Program_Chairs · 2021-01-07
**Final Decision**

**Decision:**

Accept (Poster)

**Comment:**

The paper studies nonconvex-strongly concave min-max optimization using  proximal gradient descent-ascent (GDA), assuming Kurdyka-Łojasiewicz (KŁ) condition holds. The main contribution is a novel Lyapunov function, which leads to a clean analysis. The main downsides of the paper as discussed by the reviewers are the lack of experiments and somewhat stringent assumptions needed in the analysis. Nevertheless, the paper was overall viewed favorably by the reviewers, who considered it a worthwhile contribution to the area min-max optimization.